# Weight Expansion: A New Perspective on Dropout and Generalization

**Gaojie Jin**[1], **Xinping Yi**[2*], **Pengfei Yang**[3], **Lijun Zhang**[3], **Sven Schewe**[1], **Xiaowei Huang**[1*]

[1]*Department of Computer Science, University of Liverpool*
[2]*Department of Electrical Engineering and Electronics, University of Liverpool*
[3]*State Key Laboratory of Computer Science, Institute of Software Chinese Academy of Sciences*
[*]*Corresponding author*
*g.jin3@liverpool.ac.uk,    xinping.yi@liverpool.ac.uk,    yangpf@ios.ac.cn,    zhanglj@ios.ac.cn,*
*sven.schewe@liverpool.ac.uk,    xiaowei.huang@liverpool.ac.uk*

**Reviewed on OpenReview:** `https://openreview.net/forum?id=w3z3sN1b04`

## Abstract

While dropout is known to be a successful regularization technique, insights into the mechanisms that lead to this success are still lacking. We introduce the concept of *weight expansion*, an increase in the signed volume of a parallelotope spanned by the column or row vectors of the weight covariance matrix, and show that weight expansion is an effective means of increasing the generalization in a PAC-Bayesian setting. We provide a theoretical argument that dropout leads to weight expansion and extensive empirical support for the correlation between dropout and weight expansion. To support our hypothesis that weight expansion can be regarded as an *indicator* of the enhanced generalization capability endowed by dropout, and not just as a mere by-product, we have studied other methods that achieve weight expansion (resp. contraction), and found that they generally lead to an increased (resp. decreased) generalization ability. This suggests that dropout is an attractive regularizer, because it is a computationally cheap method for obtaining weight expansion. This insight justifies the role of dropout as a regularizer, while paving the way for identifying regularizers that promise improved generalization through weight expansion.

## 1 Introduction

Research on why dropout is so effective in improving the generalization ability of neural networks has been intensive. Many intriguing phenomena induced by dropout have also been studied in this research (Gao et al., 2019; Lengerich et al., 2020; Wei et al., 2020). In particular, it has been suggested that dropout optimizes the training process (Baldi & Sadowski, 2013; Wager et al., 2013; Helmbold & Long, 2015; Kingma et al., 2015; Gal & Ghahramani, 2016; Helmbold & Long, 2017; Nalisnick et al., 2019) and that dropout regularizes the inductive bias (Cavazza et al., 2018; Mianjy et al., 2018; Mianjy & Arora, 2019; 2020).

A fundamental understanding of just how dropout achieves its success as a regularization technique will not only allow us to understand when (and why) to apply dropout, but also enable the design of new training methods. In this paper, we suggest a new measure, *weight expansion*, which both furthers our understanding and allows the development of new regularizers. Broadly speaking, the application of dropout leads to non-trivial changes in the weight covariance matrix. As the weight covariance matrix is massively parameterized and hard to comprehend, we abstract it to the signed volume of a parallelotope spanned by the vectors of the matrix—*weight volume* (normalized generalized variance (Kocherlakota & Kocherlakota, 2004)) for short. Dropout increases this weight volume. Figure 1 illustrates the weight expansion obtained by dropout and visualizes how it improves the generalization ability.

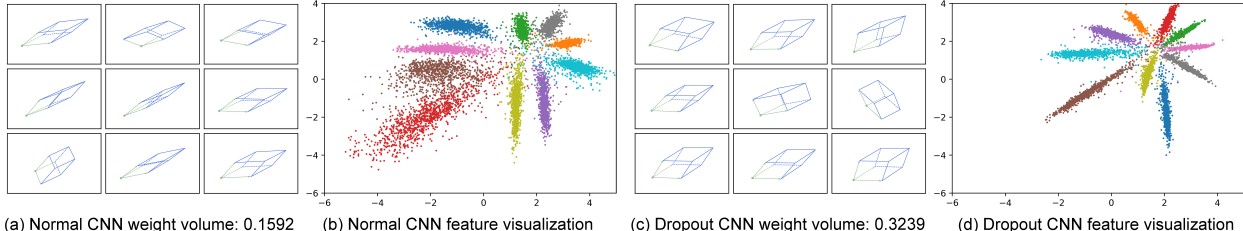

(a) Normal CNN weight volume: 0.1592  (b) Normal CNN feature visualization  (c) Dropout CNN weight volume: 0.3239  (d) Dropout CNN feature visualization

Figure 1: Visualization of weight volume and features of the last layer in a CNN on MNIST, with and without dropout during training. Plots **(b)** and **(d)** present feature visualization for normal and dropout CNN, respectively. We can see that the dropout network has clearer separation between points of different classes – a clear sign of a better generalization ability. As we will argue in this paper, this better generalization is related to the "weight expansion" phenomenon, as exhibited in plots **(a)** and **(c)**. For **(a)** and **(c)**, the weight volume of the normal CNN and the dropout CNN is 0.1592 and 0.3239, respectively. That is, *the application of dropout during training "expands" the weight volume.* For visualization purpose, we randomly select three dimensions in the weight covariance matrix and display their associated 3-dimensional parallelotope. Both **(a)** and **(c)** contains $3 \times 3 = 9$ such parallelotopes. We can see that those parallelotopes have a visibly larger volume in the dropout network. Details of the networks are provided in Appendix A.

This leads to our **hypotheses** that **(1) weight expansion reduces the generalization error** of neural networks (Section 3.1), and that **(2) dropout leads to weight expansion** (Section 3.2). We hypothesize that weight expansion can be an indicator of the increased generalization ability and dropout is 'merely' an efficient way to achieve it.

The weight volume is defined as the *normalized determinant of the weight covariance matrix (determinant of weight correlation matrix)*, which is the second-order statistics of weight vectors when treated as a multivariate random variable. For example, when the weight vector is an isotropic Gaussian, the weight volume is large and the network generalizes well. Likewise, if the weights are highly correlated, then the weight volume is small and the network does not generalize well. Technically, as a determinant, the weight volume serves as a measure of the orthogonality of the rows/columns of the weight covariance matrix. The more orthogonal the rows/columns are, the larger the weight volume is—which makes the different classes more distinguishable. Figure 1 visualizes that, with increasing weight volume, the decision boundaries of different classes become clearer. Note that, *this is different from the "weight orthogonality"* proposed in Saxe et al. (2013); Mishkin & Matas (2015); Bansal et al. (2018): Weight volume is a statistical property of weight matrices treated as random variables, while orthogonality treats weight matrices as deterministic variables (more discussions are provided in Appendix B).

In Section 3.1, we provide theoretical analysis to support *hypothesis (1)* through an extension to the PAC-Bayesian theorem, *i.e.,* weight expansion can tighten a generalization bound based on the PAC-Bayesian theorem. Then, in Section 3.2, we support *hypothesis (2)* through a one-step theoretical analysis that dropout can expand weight volume within one mini-batch.

In addition to our theoretical arguments, our empirical results also support our hypotheses. To mitigate estimation errors, we consider two independent weight volume estimation methods in Section 4, *i.e.,* sampling method and Laplace approximation. Section 5.1 demonstrates that the two methods lead to consistent results, which show that using dropout brings about weight expansion *(coherence to hypothesis (2))* and generalization-improvement *(coherence to hypothesis (1))* across data sets and networks.

Then, we embed the weight volume (a measurable quantity) to a new complexity measure that is based on our adaptation of the PAC-Bayesian bound (Section 3.1). Our experiments in Section 5.2, conducted on $3 \times 288$ network configurations, show that, comparing to existing complexity measures which do not consider weight volume, the new complexity measure is most closely correlated with generalization *(coherence to hypothesis (1))* and dropout *(coherence to hypothesis (2))* for these dropout networks with different dropout rates.

Finally, we use disentanglement noises (Section 4.1) to achieve weight expansion and further to improve generalization (or other noises to achieve contraction and further to destroy generalization, Section 5.3). This

in particular supports *hypothesis (1)* and indicates that weight expansion is a key indicator for a reduced generalization error, while dropout is a computationally inexpensive means to achieve it. (Disentanglement noises, *e.g.,* are more expensive.)

## 2 Preliminaries

**Notation.** Let $S$ be a training set with $m$ samples drawn from the input distribution $D$, and $s \in S$ a single sample. Loss, expected loss, and empirical loss are defined as $\ell(f_W(s))$, $\mathcal{L}_D(f_W) = \mathbb{E}_{s \sim D}[\ell(f_W(s))]$, and $\mathcal{L}_S(f_W) = \frac{1}{m} \sum_{s \in S}[\ell(f_W(s))]$, respectively, where $f_W(\cdot)$ is a learning model. Note that, in the above notation, we omit the label $y$ of the sample $s$, as it is clear from the context. Let $W$, $W_l$, and $w_{ij}$ be the model weight matrix, the weight matrix of $l$-th layer, and the element of the $i$-th row and the $j$-th column in $W$, respectively. Note that we omit bias for convenience. To make a clear distinction, let $\mathbf{w}_l$ be the multivariate random variable of $\text{vec}(W_l)$ (vectorization of $W_l$), and $\mathbf{w}_{ij}$ be the random variable of $w_{ij}$. $W^0$ and $W^F$ denote the weight matrix before and after training, respectively. Further, we use $W^*$ to represent the maximum-a-posteriori (MAP) estimate of $W$. Note that $W^F$ can be seen as an approximation of $W^*$ after training. We consider a feedforward neural network $f_W(\cdot)$ that takes a sample $s_0 \in S$ as input and produces an output $h_L$, after $L$ fully connected layers. At the $l$-th layer ($l = 1, \ldots, L$), the latent representation before and after the activation function is denoted as $h_l = W_l a_{l-1}$ and $a_l = \mathbf{act}_l(h_l)$, respectively, where $\mathbf{act}(\cdot)$ is the activation function.

**Node Dropout.** We focus on node dropout (Hinton et al., 2012; Srivastava et al., 2014), a popular version of dropout that randomly sets the output of a given activation layer to 0. We adopt a formal definition of node dropout from the usual Bernoulli formalism. For the layer $l$ with the output of the activation function $a_l \in \mathbb{R}^{N_l}$ and a given probability $q_l \in [0, 1)$, we sample a scaling vector $\tilde{\eta}_l \in \mathbb{R}^{N_l}$ with independently and identically distributed elements

$$(\tilde{\eta}_l)_i = \begin{cases} 0 & \text{with probability } q_l, \\ \frac{1}{1-q_l} & \text{with probability } 1 - q_l. \end{cases} \tag{1}$$

for $i = 1, \ldots, N_l$, where $N_l$ is the number of neurons at the layer $l$. With a slight abuse of notation, we denote by $\tilde{\eta}_l$ a vector of independently distributed random variables, each of which has an one mean, and also a vector with each element being either 0 or $\frac{1}{1-q_l}$ when referring to a specific realization. As such, applying node dropout, we compute the updated output after the activation as $\tilde{a}_l = \tilde{\eta}_l \odot a_l$. That is, with probability $q_l$, an element of the output with node dropout $\tilde{a}_l$ is reset to 0, and with probability $1 - q_l$ it is rescaled with a factor $\frac{1}{1-q_l}$. The index of the layer $l$ will be omitted when it is clear from the context.

## 3 Dropout and weight expansion

In this section, we first formally define the weight volume, which has been informally introduced in Figure 1. We aim to answer the following two questions: *1. Why does large weight volume promote generalization? 2. Why does dropout expand the weight volume?*

By modeling weights of neural network as random variables $\{\mathbf{w}_l\}$,[1] we have the following definition.

**Definition 3.1 (Weight Volume)** *Let $\Sigma_l = \mathbb{E}[(\mathbf{w}_l - \mathbb{E}(\mathbf{w}_l))(\mathbf{w}_l - \mathbb{E}(\mathbf{w}_l))^\mathsf{T}]$ be the weight covariance matrix in a neural network. The weight volume (also called as normalized generalized variance or determinant of weight correlation matrix) is defined as*

$$\text{vol}(\mathbf{w}_l) \triangleq \frac{\det(\Sigma_l)}{\prod_i [\Sigma_l]_{ii}} \in [0, 1], \tag{2}$$

---

[1]The modeling of neural network weights as random variables has been widely adopted in the literature for mutual information estimation (Xu & Raginsky, 2017; Negrea et al., 2019), PAC-Bayesian analysis (McAllester, 1999; Langford & Caruana, 2002; Dziugaite & Roy, 2017), to name just a few. Under DNN PAC-Bayesian framework, previous works (sharpness PAC-Bayes in Jiang et al. (2019), Flatness PAC-Bayes in Dziugaite et al. (2020b)) assume Gaussian random weights scatter around the DNN, with the condition that their generalization performance is similar to the DNN's, to compute the bound in practice. The networks parameterized by these random weights can be considered to lie in the same local minima as they are close and have similar generalization performance. We also follow this assumption and estimate weight volume in Section 4.

*where the denominator is the product of the diagonal elements in $\Sigma_l$.*

Intuitively, the weight volume comes from the geometric interpretation of the determinant of the matrix $\det(\Sigma_l)$, with normalization over the diagonal elements of the matrix. Because $\Sigma_l$ is a covariance matrix and thus positive semi-definite, we have $\text{vol}(\mathbf{w}_l) \in [0,1]$ since $0 \le \det(\Sigma_l) \le \prod_i [\Sigma_l]_{ii}$. We note that $\text{vol}(\mathbf{w}_l) = 0$ implies that the the weight covariance matrix does not have full rank, suggesting that some weights are completely determined by the others, while $\text{vol}(\mathbf{w}_l) = 1$ suggests that the weights $\mathbf{w}_l$ are uncorrelated. The larger $\text{vol}(\mathbf{w}_l)$ is, the more dispersed are the weights, which means random weights lie in a more spherical (i.e., flattened) local minima.

Before presenting empirical results, we first provide theoretical support for our first hypothesis by adapting the PAC-Bayesian theorem (McAllester, 1999; Dziugaite & Roy, 2017) (Section 3.1): We show that a larger weight volume leads to a tighter bound for the generalization error. In Section 3.2, we argue that dropout can expand weight volume through the theoretical derivation.

### 3.1 Why does large weight volume promote generalization?

First, we support *hypothesis (1)* on the relevance of weight expansion through an extension of the PAC-Bayesian theorem, and then verify it empirically in Section 5. The PAC-Bayesian framework, developed in McAllester (1999); Neyshabur et al. (2017); Dziugaite & Roy (2017), connects weights with generalization by establishing an upper bound on the generalization error with respect to the Kullback-Leibler divergence ($D_{\text{KL}}$) between the posterior distribution $Q$ and the prior distribution $P$ of the weights. In the following, we extend this framework to work with $\text{vol}(\mathbf{w}_l)$.

Let $P$ be a prior and $Q$ be a posterior over the weight space. For any $\delta > 0$, with probability $1 - \delta$ over the draw of the input space, we have (Dziugaite & Roy, 2017)

$$\mathbb{E}_{\mathbf{w} \sim Q}[\mathcal{L}_D(f_\mathbf{w})] \le \mathbb{E}_{\mathbf{w} \sim Q}[\mathcal{L}_S(f_\mathbf{w})] + \sqrt{\frac{D_{\text{KL}}(Q||P) + \ln \frac{m}{\delta}}{2(m-1)}}. \tag{3}$$

Given a learning setting, Dziugaite & Roy (2017); Neyshabur et al. (2017); Jiang et al. (2019) assume $P = \mathcal{N}(\mu_P, \Sigma_P)$ and $Q = \mathcal{N}(\mu_Q, \Sigma_Q)$. Thus, $D_{\text{KL}}(Q||P)$ can be simplified to

$$D_{\text{KL}}(Q||P) = \frac{1}{2}\Big(\text{tr}(\Sigma_P^{-1}\Sigma_Q) - k + (\mu_P - \mu_Q)^\intercal (\Sigma_P)^{-1}(\mu_P - \mu_Q) + \ln \frac{\det(\Sigma_P)}{\det(\Sigma_Q)}\Big), \tag{4}$$

where $k$ is the dimension of $\mathbf{w}$, tr denotes the trace, and det denotes the determinant. The derivation is given in Appendix D. To simplify the analysis, Neyshabur et al. (2017; 2018a) instantiate the prior $P$ to be a $\text{vec}(W^0)$ (or $\mathbf{0}$) mean and $\sigma^2$ variance Gaussian distribution, and assume that the posterior $Q$ to also be a $\text{vec}(W^F)$ mean spherical Gaussian with variance $\sigma^2$ in each direction.

We *relax their assumption* by letting $Q$ be a non-spherical Gaussian (the off-diagonal correlations for same layer are not 0, whereas those for different layers are 0), while retaining the assumption on the $\sigma^2$ variance in every direction (but allowing arbitrary covariance between them). This retains both $\text{tr}(\Sigma_P^{-1}\Sigma_Q) = k$ and $\prod_i [\Sigma_{l,P}]_{ii} = \prod_i [\Sigma_{l,Q}]_{ii}$ for all $l$. Then, we get the following Lemma.

**Lemma 3.1 (Weight expansion reduces $D_{\text{KL}}(Q||P)$)** *Let $P = \mathcal{N}(\text{vec}(W^0), \Sigma_P)$, $\Sigma_P = \sigma^2 \cdot I$, $Q = \mathcal{N}(\text{vec}(W^F), \Sigma_Q)$, the diagonal elements of $\Sigma_Q$ be $\sigma^2$, and the covariances for different layers in $\Sigma_Q$ be 0. Thus, we get*

$$D_{\text{KL}}(Q||P) = \frac{1}{2}\sum_l \Big(\frac{||W_l^F - W_l^0||_F^2}{\sigma^2} + \ln \frac{1}{\text{vol}(\mathbf{w}_l)}\Big). \tag{5}$$

Details of the derivation are provided in Appendix D. Note that Lemma 3.1 is a theoretical analysis from prior to posterior, in which we regard the whole training procedure as a single step. From Equations (3) and (5), we notice that the PAC-Bayesian upper bound of the generalization error becomes smaller when $\text{vol}(\mathbf{w}_l)$ increases, which is also demonstrated by wide experiments in Section 5.

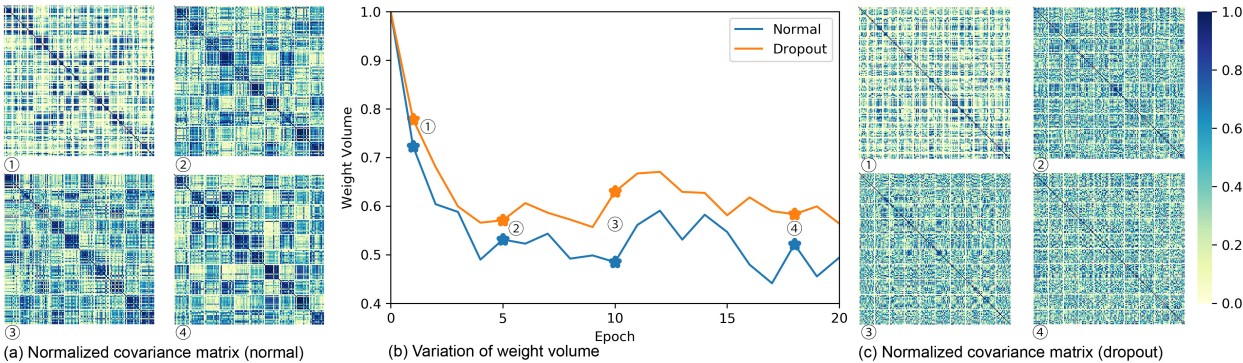

Figure 2: We train two small NNs (64-32-16-10 network with/without dropout) on MNIST. **(b)** shows the changes in weight volume along with epochs. There are four sampling points (①②③④) to track the **absolute** normalized covariance matrix (correlation matrix) of normal gradient updates **(a)** and dropout gradient updates **(c)**. The results for VGG16 on CIFAR-10 are provided in Appendix C.

### 3.2 Why does dropout expand the weight volume?

In the above PAC-Bayesian framework, we assume the (non-spherical) Gaussian distributed weights and connect weight volume with generalization. In this subsection, to connect weight volume with dropout through correlation (of gradient updates and weights), and further to support *hypothesis (2)*, we consider *the arbitrarily distributed* (with finite variance) gradient updates of $l$-th layer, $\Delta_l$ for a normal network and $\tilde{\Delta}_l$ for the corresponding dropout network (all learning settings are same except for dropout). That is, let $\text{vec}(\Delta_l)$ and $\text{vec}(\tilde{\Delta}_l')$ be i.i.d. and follow an arbitrary distribution $\mathcal{M}_\Delta(\mu_{\Delta_l}, \Sigma_{\Delta_l})$, and dropout noise $\tilde{\eta}_l$ be applied to the gradient updates through

$$\text{vec}(\tilde{\Delta}_l) = (\mathbf{1} \otimes \tilde{\eta}_l) \odot \text{vec}(\tilde{\Delta}_l'),$$

where the one vector $\mathbf{1} \in \mathbb{R}^{N_{l-1}}$ and the random vector $\mathbf{1} \otimes \tilde{\eta}_l \in \mathbb{R}^{N_{l-1}N_l}$ with $\otimes$ being Kronecker product. *As we aim to study the impact of $\tilde{\eta}_l$ on the correlation of gradient updates, we assume $\text{vec}(\Delta_l)$ and $\text{vec}(\tilde{\Delta}_l')$ are i.i.d. to simplify the correlation analysis.* Thus, with a slight abuse of notation, denote $(\mathbf{1} \otimes \tilde{\eta}_l) \odot \text{vec}(\tilde{\Delta}_l')$ as $\tilde{\eta}_l$ to be applied on gradient updates of corresponding neurons for notational convenience, we can now obtain Lemma 3.2.

**Lemma 3.2 (Dropout reduces correlation of gradient updates)** *Let $\Delta_{ij}$ and $\Delta_{i'j'}$ be the gradient updates for $\mathbf{w}_{ij}$ and $\mathbf{w}_{i'j'}$ of some layer in a normal network, where $i \neq i'$, $\tilde{\Delta}_{ij}$ and $\tilde{\Delta}_{i'j'}$ be the gradient updates for the same setting dropout network. Let $\text{vec}(\Delta)$, $\text{vec}(\tilde{\Delta}')$ be i.i.d. and follow an arbitrary distribution $\mathcal{M}_\Delta(\mu_\Delta, \Sigma_\Delta)$, $\text{vec}(\tilde{\Delta}) = (\mathbf{1} \otimes \tilde{\eta}) \odot \text{vec}(\tilde{\Delta}')$. As dropout noises for different nodes (units) are independent, the (absolute) correlation between $\tilde{\Delta}_{ij}$ and $\tilde{\Delta}_{i'j'}$ in a dropout network can be upper bounded by that in the normally trained network with the same setting, i.e.,*

$$\left|\rho_{\tilde{\Delta}_{ij}, \tilde{\Delta}_{i'j'}}\right| \leq (1-q)\left|\rho_{\Delta_{ij}, \Delta_{i'j'}}\right|. \tag{6}$$

Details of the proof are given in Appendix E. The equality holds if, and only if, $q = 0$ or $\rho_{\Delta_{ij}, \Delta_{i'j'}} = 0$. Lemma 3.2 shows that the dropout noise decreases correlation among gradient updates, which is also shown by experiments in Figure 2 and Appendix C. That is, the correlation among gradient updates in a normal network (Figure 2a) is obviously larger than the correlation in a dropout network (Figure 2c). Next, Lemma 3.2 leads to the following lemma.

**Lemma 3.3 (Dropout reduces correlation of updated weights)** *Given the assumptions in Lemma 3.2 and Gaussian weights, we get that the (absolute) correlation of updated weights in a dropout network is upper bounded by that in the normal network. That is,*

$$\left|\rho_{\mathbf{w}_{ij}+\tilde{\Delta}_{ij}, \mathbf{w}_{i'j'}+\tilde{\Delta}_{i'j'}}\right| \leq \left|\rho_{\mathbf{w}_{ij}+\Delta_{ij}, \mathbf{w}_{i'j'}+\Delta_{i'j'}}\right|. \tag{7}$$

---

**Algorithm 1** Sampling Method.

---

**Input:** Convergent network $f_W$, data set $S$, iterations $n$, Convergent network loss $\mathcal{L}(f_W)$

▶$W' \leftarrow W$

▶**for** $i = 1$ to $n$ **do**

    $\text{vec}(W') \leftarrow \text{vec}(W') + \mathcal{N}(\mathbf{0}, 0.01 \cdot I)$

    $W'$ is updated by $f_{W'}(S)$ with 1 epoch, 0.0001 learning rate

    Collect $W'$ as a sample if $|\mathcal{L}(f_{W'}) - \mathcal{L}(f_W)| \le \epsilon$

▶**end for**

---

The equality holds if and only if $q = 0$ or $\text{Cov}(\mathbf{w}_{ij} + \Delta_{ij}, \mathbf{w}_{i'j'} + \Delta_{i'j'}) = 0$, and the full proof is given in Appendix F. It is clear that weight volume (equals the determinant of weight correlation matrix) is one of measures of correlation among weights (refer to Figure 1), and small correlation can lead to large weight volume. We also provide more discussions about the relationship between correlation and weight volume in Appendix G. Consistent with Lemma 3.3, we show in addition, that gradient updates with large correlation (Figure 2a, normal network) can lead to a large correlation of weights—and thus to a small weight volume (Figure 2b), while gradient updates with small correlation (Figure 2c, dropout network) can lead to a small correlation of weights—and thus to a large weight volume.

Note that, as the correlation during the whole training procedure is too complicated to be analyzed, Lemmas 3.2 and 3.3 adopt one-step theoretical argument (with one mini-batch update). With the same pre-condition, we prove that dropout noise can reduce the correlation among weights. Based on the one-step theoretical result, our results in Figure 2 empirically show that this phenomenon is extensible to the whole training procedure.

## 4 Weight volume approximation

To mitigate estimation errors, we use two methods to estimate the weight volume. One is the sampling method, and the other is Laplace approximation (Botev et al., 2017; Ritter et al., 2018) of neural networks, which models dropout as a noise in the Hessian matrix. Laplace approximation enables us to generalize dropout to other disentanglement noises (Section 4.1). Both sampling method and Laplace approximation follow the assumption in Footnote 1. Note that some of other popular regularizers, *e.g.,* weight decay and batch normalization, *improve generalization not through weight expansion* (more discussions are provided in Appendix H).

**Sampling Method.** First, we use a sharpness-like method (Keskar et al., 2016; Jiang et al., 2019) to get a set of weight samples drawn from $(W + \mathbf{U})$ such that $|\mathcal{L}(f_{W+U}) - \mathcal{L}(f_W)| \le \epsilon$, where $\text{vec}(\mathbf{U}) \sim \mathcal{N}(0, \sigma_{\mathbf{U}}^2 I)$ is a multivariate random variable obeys zero mean Gaussian with $\sigma_{\mathbf{U}} = 0.1$. To get the samples from the posterior distribution steadily and fastly, we train the convergent network with learning rate 0.0001 and noise $\mathbf{U}$, then collect corresponding samples. We also provide the pseudo-code for sampling method in Algorithm 1. As the samples are stabilized at training loss but with different weights, we can treat them as the samples from same posterior distribution. The covariance between $\mathbf{w}_{ij}$ and $\mathbf{w}_{i'j'}$ can then be computed by $\text{Cov}(\mathbf{w}_{ij}, \mathbf{w}_{i'j'}) = \mathbb{E}[(\mathbf{w}_{ij} - \mathbb{E}[\mathbf{w}_{ij}])(\mathbf{w}_{i'j'} - \mathbb{E}[\mathbf{w}_{i'j'}])]$. Finally, we can estimate $\text{vol}(\mathbf{w})$ according to Equation (2).

**Laplace Approximation.** Laplace approximation has been used in posterior estimation in Bayesian inference (Bishop, 2006; Ritter et al., 2018). It aims to approximate the posterior distribution $p(\mathbf{w}|S)$ by a Gaussian distribution, based on the second-order Taylor approximation of the ln posterior around its MAP estimate. Specifically, for layer $l$ and given weights with an MAP estimate $W_l^*$ on $S$, we have

$$\ln p(\mathbf{w}_l|S) \approx \ln p\Big(\text{vec}(W_l^*)|S\Big) - \frac{1}{2}\Big(\mathbf{w}_l - \text{vec}(W_l^*)\Big)^{\mathsf{T}} \Sigma_l^{-1} \Big(\mathbf{w}_l - \text{vec}(W_l^*)\Big), \tag{8}$$

where $\Sigma_l^{-1} = \mathbb{E}_s[H_l]$ is the expectation of the Hessian matrix over input data sample $s$, such that the Hessian matrix $H_l$ is given by $H_l = \frac{\partial^2 \ell(f_W(s))}{\partial \text{vec}(W_l)\partial \text{vec}(W_l)}$.

---

**Algorithm 2** Gradient update with disentanglement noise.

---

**Input:** minibatch $\{s_i\}_{i=1}^n$, activation noise $\eta_a$, node loss noise $\eta_h$, noise strengths $\lambda_1$, $\lambda_2$.

▶ Forward Propagation: apply noise $\lambda_1 \cdot \eta_a$ to activations.

▶ Back Propagation: apply noise $\lambda_2 \cdot \eta_h$ to node loss.

▶ Calculate $g = \frac{1}{n} \sum_{i=1}^n \nabla_W \big( \ell(f_W(s_i)) \big)$ by considering both noises as indicated above.

▶ Use $g$ for the optimization algorithm.

---

It is worth noting that the gradient is zero around the MAP estimate $W^*$, so the first-order Taylor polynomial is inexistent. Taking a closer look at Equation 8, one can find that its right hand side is exactly the logarithm of the probability density function of a Gaussian distributed multivariate random variable with mean $\text{vec}(W_l^*)$ and covariance $\Sigma_l$, *i.e.*, $\mathbf{w}_l \sim \mathcal{N}(\text{vec}(W_l^*), \Sigma_l)$, where $\Sigma_l$ can be viewed as the covariance matrix of $\mathbf{w}_l$.

Laplace approximation suggests that it is possible to estimate $\Sigma_l$ through the inverse of the Hessian matrix, because $\Sigma_l^{-1} = \mathbb{E}_s[H_l]$. Recently, Botev et al. (2017); Ritter et al. (2018) have leveraged insights from second-order optimization for neural networks to construct a Kronecker factored Laplace approximation. Differently from the classical second-order methods (Battiti, 1992; Shepherd, 2012), which suffer from high computational costs for deep neural networks, it takes advantage of the fact that Hessian matrices at the $l$-th layer can be Kronecker factored as explained in Martens & Grosse (2015); Botev et al. (2017). *I.e.*,

$$H_l = \underbrace{a_{l-1} a_{l-1}^\intercal}_{\mathcal{A}_{l-1}} \otimes \underbrace{\frac{\partial^2 \ell(f_W(s))}{\partial h_l \partial h_l}}_{\mathcal{H}_l} = \mathcal{A}_{l-1} \otimes \mathcal{H}_l, \tag{9}$$

where $\mathcal{A}_{l-1} \in \mathbb{R}^{N_{l-1} \times N_{l-1}}$ indicates the subspace spanned by the post-activation of the previous layer, and $\mathcal{H}_l \in \mathbb{R}^{N_l \times N_l}$ is the Hessian matrix of the loss with respect to the pre-activation of the current layer, with $N_{l-1}$ and $N_l$ being the number of neurons at the $(l-1)$-th and $l$-th layer, respectively. Further, through the derivation in Appendix I, we can estimate $\text{vol}(\mathbf{w}_l)$ efficiently by having

$$\det(\Sigma_l) \approx \det \big( (\mathbb{E}_s[\mathcal{A}_{l-1}])^{-1} \big)^{N_l} \cdot \det \big( (\mathbb{E}_s[\mathcal{H}_l])^{-1} \big)^{N_{l-1}} \tag{10}$$

and

$$\prod_i [\Sigma_l]_{ii} \approx \big( \prod_c \big[ (\mathbb{E}_s[\mathcal{A}_{l-1}])^{-1} \big]_{cc} \big)^{N_l} \cdot \big( \prod_d \big[ (\mathbb{E}_s[\mathcal{H}_l])^{-1} \big]_{dd} \big)^{N_{l-1}}. \tag{11}$$

Note that, for the case with dropout, dropout noise $\tilde{\eta}$ is taken into account to compute $\mathbb{E}_s[\mathcal{A}_{l-1}]$ and $\mathbb{E}_s[\mathcal{H}_l]$. That is, $\mathbb{E}_s[\tilde{\mathcal{A}}_{l-1}] = \mathbb{E}_{s,\tilde{\eta}_l}[\tilde{a}_{l-1}(\tilde{a}_{l-1})^\intercal]$ and $\mathbb{E}_s[\tilde{\mathcal{H}}_l] = \mathbb{E}_{s,\tilde{\eta}_l}[\frac{\partial^2 \ell(f_W(s))}{\partial \tilde{h}_l \partial \tilde{h}_l}]$, where $\tilde{a}_{l-1}$ and $\tilde{h}_l$ are the activation and pre-activation of the corresponding layer in a dropout network.

### 4.1 Other disentanglement noise

Whilst introducing dropout noise can expand weight volume (as shown in the experiments of Sections 5.1, 5.2), it is interesting to know (1) whether there are other noises that can lead to *weight expansion*, and, if so, whether they also lead to better generalization performance, and (2) whether there are noises that can lead to *weight contraction*, and, if so, whether those noises lead to worse generalization performance.

As indicated in Equation (9) and Appendix I, the weight covariance matrix can be approximated by the Kronecker product of $(\mathbb{E}[\mathcal{A}_{l-1}])^{-1}$ and $(\mathbb{E}[\mathcal{H}_l])^{-1}$. Disentanglement noises, as an attempt, can be injected into either term during the gradient update procedure to increase $\det(\mathbb{E}[\mathcal{A}_{l-1}])^{-1}$ and $\det(\mathbb{E}[\mathcal{H}_l])^{-1}$ for the purpose of weight expansion. The first option is to inject $\eta_a$ to activations during forward propagation, as follows.

**Activation Reverse Noise $\eta_a$ to increase** $\det(\mathbb{E}[\mathcal{A}])^{-1}$. We define $\eta_a$ as $\eta_a \sim \mathcal{N}(\mu, r(\Sigma_a))$, where $\Sigma_a = (\mathbb{E}[\mathcal{A}])^{-1}$ and $r(\Sigma)$ is to reverse the signs of off-diagonal elements of the matrix $\Sigma$.

The second option is to inject $\eta_h$ into the node loss during backpropagation, as follows.

Table 1: Weight volume on VGG networks.

| | Method | VGG11 | VGG11(dropout) | VGG16 | VGG16(dropout) | VGG19 | VGG19(dropout) |
|---|---|---|---|---|---|---|---|
| CIFAR-10 | Sampling | 0.06±0.02 | **0.13±0.02** | 0.05±0.02 | **0.12±0.02** | 0.04±0.02 | **0.12±0.02** |
| | Laplace | 0.0568 | **0.1523** | 0.0475 | **0.1397** | 0.0453 | **0.1443** |
| | Generalization Gap | 0.8030 | **0.5215** | 0.8698 | **0.2996** | 1.1087 | **0.1055** |
| CIFAR-100 | Sampling | 0.07±0.02 | **0.14±0.02** | 0.06±0.02 | **0.14±0.02** | 0.04±0.02 | **0.12±0.02** |
| | Laplace | 0.0537 | **0.1578** | 0.0409 | **0.1620** | 0.0506 | **0.1409** |
| | Generalization Gap | 3.1208 | **2.1635** | 3.7541 | **1.2714** | 4.4787 | **0.4373** |

**Node Loss Reverse Noise $\eta_h$ to increase** $\det(\mathbb{E}[\mathcal{H}])^{-1}$**.** We define $\eta_h$ as $\eta_h \sim \mathcal{N}(\mu, r(\Sigma_h))$, where $\Sigma_h = (\mathbb{E}[\mathcal{H}])^{-1}$.

Algorithm 2 presents our simple and effective way to obtain disentanglement noise, where $\lambda_1$ and $\lambda_2$ are hyper-parameters to balance the strengths of $\eta_a$ and $\eta_h$. In the experiments in Section 5.3, we will assign expansion noises and contraction noises according to their impact on the weight volume, where expansion noises are obtained from Algorithm 2 and contraction noises are obtained from stochastic noises on weights.

## 5 Experiments

We have conducted a number of experiments to test our *hypotheses* that *(1) weight expansion helps reduce the generalization error* and *(2) dropout increases the weight volume.* More than 900 networks are trained. First, we confirm the role of weight expansion in connecting dropout and generalization by estimating the weight volume with two independent methods (Section 5.1). Second, we extend the PAC-Bayesian theorem with the awareness to weight volume, and show that the new complexity measure can significantly improve the prediction of the generalization error for dropout networks (Section 5.2). Finally, we consider other noises that can lead to either weight expansion or weight contraction, and study their respective impact on the generalization performance (Section 5.3). All empirical results are both significant and persistent across the models and data sets we work with, and support either hypothesis (1) or hypothesis (2) or both. In our experiments, we consider VGG-like models (Simonyan & Zisserman, 2014) and AlexNet-like models (Krizhevsky et al., 2012) for CIFAR-10/100 (Krizhevsky et al., 2009), ImageNet-32 (Deng et al., 2009), and SVHN (Netzer et al., 2011). For sampling method (and sharpness method), we default to the settings of $\epsilon = 0.05$ for CIFAR-10/SVHN, and $\epsilon = 0.1$ for CIFAR-100/ImageNet-32. Sampling method takes about 1,500s to collect samples and estimate weight volume for each model, and Laplace Approximation takes about 300s for each model.

### 5.1 Measuring weight volume

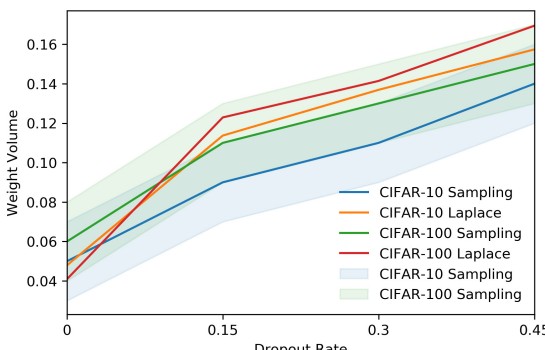

Figure 3: Weight volume w.r.t. dropout rate.

First experiment studies the correlation between weight volume and the application of dropout during training. Weight volume is hard to measure exactly, so we use two different, yet independent, estimation methods—Laplace approximation and the sampling method as in Section 4—to improve our confidence in the results and robustly test our hypotheses.

Table 1 presents the estimated weight volume and the generalization gap (loss gap) on several VGG networks on CIFAR-10/100. We see that dropout leads, across all models, to a persistent and significant increase of the weight volume (*hypothesis (2)*). At the same time, the generalization error is reduced (*hypothesis (1)*). We default to the setting of weight decay with 0 in the experiments of Table 1, Figure 3, and also present similar results with weight decay 0.0005 (and more results on ImageNet-32) in Appendix J, text classification experiments in Appendix M.1.

Table 2: Mutual Information of complexity measures on CIFAR-10.

| Complexity | VGG | | | | AlexNet | | | |
|---|---|---|---|---|---|---|---|---|
| Measure | Dropout | GG($|\omega|$=2) | GG($|\omega|$=1) | GG($|\omega|$=0) | Dropout | GG($|\omega|$=2) | GG($|\omega|$=1) | GG($|\omega|$=0) |
| Frob Distance | 0.0233 | 0.0243 | 0.0127 | 0.0039 | 0.0644 | 0.0289 | 0.0159 | 0.0094 |
| Spectral Norm | 0.0231 | 0.0274 | 0.0139 | 0.0001 | 0.1046 | 0.0837 | 0.0832 | 0.0786 |
| Parameter Norm | 0.0535 | 0.0252 | 0.0122 | 0.0008 | 0.0567 | 0.0814 | 0.0793 | 0.0744 |
| Path Norm | 0.0697 | 0.0129 | 0.0053 | 0.0027 | 0.1530 | 0.0425 | 0.0307 | 0.0177 |
| Sharpness $\alpha'$ | 0.1605 | 0.0127 | 0.0053 | 0.0001 | 0.1583 | 0.0877 | 0.0639 | 0.0418 |
| PAC-Sharpness | 0.0797 | 0.0200 | 0.0109 | 0.0023 | 0.1277 | 0.0552 | 0.0324 | 0.0097 |
| **PAC-S(Laplace)** | 0.5697 | 0.0837 | **0.0623** | **0.0446** | **0.7325** | **0.1248** | **0.1123** | **0.1052** |
| **PAC-S(Sampling)** | **0.6606** | **0.0857** | **0.0623** | 0.0433 | 0.5917 | 0.1014 | 0.0921 | 0.0847 |

After confirming the qualitative correlation between weight volume and dropout, we consider the quantitative effect of the dropout rate on the weight volume. As Figure 3 shows, for the 4 VGG16 models with dropout rate 0.0, 0.15, 0.3, 0.45, respectively, we see a clear trend on CIFAR-10/100 that the weight volume increases with the dropout rate (*hypothesis (2)*). The generalization gap is also monotonically reduced (CIFAR-10: 0.86, 0.51, 0.30, 0.16; CIFAR-100: 3.75, 2.14, 1.07, 0.39) (*hypothesis (1)*).

In summary, these experiments provide evidence to the correlation between dropout, weight expansion, and generalization. This raises two follow-up questions: (1) can the weight volume serve as a proxy for the generalization gap? and (2) can techniques other than dropout exhibit a similar behavior? Sections 5.2, 5.3 answer these questions in the affirmative, shedding light on the nature of the correlation between these concepts.

**Limitations.** There is a problem when dropout rate is large (*i.e.*, close to 1), as it will seriously reduce the training accuracy. Although in this case the generalization gap will be small and weight volume will be large, the performance of the model is so bad that the dropout does not actually improve the training. Thus, we only consider suitable dropout rate ($[0, 0.45]$) in our experiments.

## 5.2 Performance of the new complexity measure

A complexity measure can predict generalization gap without resorting to a test set. Inspired by the experiments in Jiang et al. (2019), we have trained 288*3=864 dropout networks (with different rates: 0, 0.15, 0.3) on CIFAR-10 for VGG-like and AlexNet-like structures and CIFAR-100 for VGG-like structures to compare our new complexity measure—which incorporates the weight volume—with a set of existing complexity measures, including PAC-Sharpness, which was shown in Jiang et al. (2019) to perform better overall than many others. Following Jiang et al. (2019), we use mutual information (**MI**) to measure the correlation between various complexity measures, and dropout and generalization gap (**GG**, defined as $\mathcal{L}_{S_{test}} - \mathcal{L}_{S_{train}}$). The details of experimental settings (*e.g.,* **GG**($|\omega|$=2)) and **MI** are given in Appendix K.1.

We compare our new complexity measures (Equation (5), with Laplace approximation and sampling method respectively) with some existing ones, including Frobenius (Frob) Distance, Parameter Norm (Dziugaite & Roy, 2017; Neyshabur et al., 2018b; Nagarajan & Kolter, 2019), Spectral Norm (Yoshida & Miyato, 2017), Path Norm (Neyshabur et al., 2015a;b), Sharpness $\alpha'$, and PAC-Sharpness (Keskar et al., 2016; Neyshabur et al., 2017; Pitas et al., 2017). Among them, PAC-Sharpness is suggested in Jiang et al. (2019) as the the most promising one in their experiments. The variance $\sigma^2$ of our methods (PAC-S(Laplace), PAC-S(Sampling)) and PAC-Sharpness is estimated by sharpness method, *i.e.,* let vec($U'$) $\sim \mathcal{N}(\mathbf{0}, \sigma'^2 I)$ be a sample from a zero mean Gaussian, where $\sigma$ is chosen to be the largest number such that $\max_{\sigma' \leq \sigma} |\mathcal{L}(f_{W+U'}) - \mathcal{L}(f_W)| \leq \epsilon$ (Jiang et al., 2019). Note that, the *perturbed* models are used to estimate weight volume (by using sampling method) and sharpness PAC-Bayes sigma. Then, the complexity measures with these estimates are further used to predict the generalization of *unperturbed* models.

Table 2 presents the **MI** as described above on CIFAR-10 for VGG-like models and AlexNet-like models. Details of the above complexity measures are given in Appendix K.2. More empirical results on CIFAR-100 are given in Appendix K.3. We can see that, for most cases concerning the generalization error, i.e., $|\omega| = 0, 1, 2$ (fix 0, 1, 2 hyper-parameters respectively, and compute corresponding expected **MI** over sub-spaces, details are shown in Appendix K.1), our new measures, PAC-S(Laplace) and PAC-S(Sampling), achieve the highest

Figure 4: **Disentanglement noise vs. dropout.** We have trained 2 VGG16 networks (**Left**) and 2 AlexNet (**Right**) on ImageNet-32 and CIFAR-100 respectively, with normal training, dropout training, disentanglement noise training (volume expansion) and stochastic noise training (volume contraction). For each data set, we present their test losses in different plots. More experiments on the CIFAR-10, SVHN are given in Appendix L.

**MI** with **GG** among all complexity measures. On AlexNet, the **MI** of other complexity measures increase, but our new measures still perform the best. Generally, the **MI** with **GG** for our measures are significantly higher than those for others (which supports *hypothesis (1)*), because our complexity measures can predict generalization-improvement from weight expansion, which is mainly caused by dropout in our network configurations. Meanwhile, *hypothesis (2)* is also empirically verified as our complexity measures are most closely correlated with dropout (see **MI** between our complexity measures and dropout). In summary, the above results demonstrate that weight expansion can be regarded as an indicator of the enhanced generalization capability (see **MI** with **GG**) endowed by dropout (see **MI** with dropout).

### 5.3   Disentanglement noise other than dropout

The experiments in Sections 5.1, 5.2 provide support to the *hypotheses (1) and (2)* that dropout, weight expansion, and generalization are correlated, with an indication that weight expansion is the proxy to the connection between dropout and the generalization error. Our final experiment is designed to test this observation by considering the replacement of dropout.

We consider the disentanglement noise as presented in Algorithm 2, and train a set of VGG16 networks and AlexNet on CIFAR-10, CIFAR-100, Imagenet-32 and SVHN, with normal training, dropout training, disentanglement noise training (volume expansion), and stochastic noise training (volume contraction), respectively. As stochastic noises on weights may lead to either weight volume contraction or leave it unchanged, we collect the contraction cases for our experiments.

Figure 4 shows that volume expansion improves generalization performance, similar to dropout. Likewise, volume contraction leads to a worse generalization performance. This relation, which further supports *hypothesis (1)*, is persistent across the examined networks and data sets. This suggests that the weight expansion, instead of dropout, can serve as a key indicator of good generalization performance, while dropout provides one method to implement volume expansion. Disentanglement noises may be one of valid replacements for dropout.

## 6   Discussion

### 6.1   Related work about dropout

Optimizing the training process to improve generalization has been a hot research area, among which a lot of work studies it with respect to dropout. For example, Wager et al. (2013) considers training procedure with adaptive dropout; Gal & Ghahramani (2016) interprets dropout with a spike-and-slab distribution by variational approximation; and Kingma et al. (2015) proposes a variational dropout, which is an extension of the Gaussian dropout in which the dropout rates are learned during training. By taking into account continuous distributions and Bernoulli noise (i.e. dropout), Nalisnick et al. (2019) provides a framework for comprehending multiplicative noise in neural networks. Instead of considering directly, and mostly

qualitatively, how dropout improves the training as above, we understand dropout and generalization through a measurable quantity – weight volume.

Recent research has started to provide a narrative understanding of dropout's performance (Zhang et al., 2021), and providing variational dropout (Wang & Manning, 2013; Gal & Ghahramani, 2016; Gal et al., 2017; Achille & Soatto, 2018; Fan et al., 2020; Lee et al., 2020; Pham & Le, 2021; Fan et al., 2021). For example, Gao et al. (2019) suggests that dropout can be separated into forward dropout (for the forward pass) and backward dropout (for the backward pass), and different dropout ratios might be needed to fully utilize the benefit of these two passes. Wei et al. (2020) empirically and theoretically explains the effectiveness of dropout on LSTM and transformer architectures from two entangled sources of regularization. It is unclear if this also holds for convolutional networks. Lengerich et al. (2020) considers input and activation dropout, suggesting that dropout reduces spurious high order interaction between input features. Different from these, we consider explaining dropout through the rigorously defined concept of weight expansion.

Providing a theoretical analysis to understand dropout from the perspective of regularizing the inductive bias, Cavazza et al. (2018) considers dropout for matrix factorization, and shows that dropout achieves the global minimum of a convex approximation problem with (squared) nuclear norm regularization. This is extended to deep linear network (Mianjy et al., 2018), deep linear network with squared loss (Mianjy & Arora, 2019), and DNN with only one last dropout layer (Arora et al., 2019). Mianjy & Arora (2020) presents precise iteration complexity results for dropout training in two-layer ReLU networks, using the logistic loss when the data distribution is separable in the reproducing kernel Hilbert space. However, it is unclear how to extend this framework to large networks. From another perspective, we present theoretical analysis that dropout increases weight volume to improve generalization.

## 6.2 Related work about other generalization factors

Other than dropout, there are works studying how other factors affect the generalization of neural networks (Chatterji et al., 2020; Dziugaite et al., 2020a; Wu et al., 2020; Yang et al., 2020; Harutyunyan et al., 2020; Haghifam et al., 2020), and the relation between gradient noises and generalization (Keskar & Socher, 2017; Jastrzębski et al., 2017; Smith & Le, 2017; Xing et al., 2018; Chaudhari & Soatto, 2018; Jastrzębski et al., 2018; Li et al., 2019). For example, Wen et al. (2019) adds diagonal Fisher noise to large-batch gradient updates, making it match the behavior of small-batch SGD. Zhu et al. (2019) investigates unbiased noise in a generic type of gradient based optimization dynamics that combines SGD and standard Langevin dynamics. Menon et al. (2020) demonstrates that in some circumstances, post-hoc processing can enhance the performance of overparameterized models on under-represented subgroups. Moreover, Jin et al. (2020) studies the relation between generalization and a definition of correlation on weight matrices. There was also a NeurIPS 2020 Competition on "Predicting Generalization in Deep Learning" (Jiang et al., 2020; Lassance et al., 2020; Natekar & Sharma, 2020). To study generalization, Huang et al. (2020) represents the Fisher information matrix of DNNs as block diagonal matrix, Huang et al. (2021) lets the filter of convolutional layer obey a block diagonal covariance matrix Gaussian. In this work, we introduce a novel factor weight volume based on random weights, which also has a close relationship with generalization.

## 6.3 Related work about PAC-Bayes

PAC-Bayes is a general framework to efficiently study generalization for numerous machine learning models. Since the invention of the PAC-Bayes (McAllester, 1999), there have been a number of canonical works developed in the past decades, *e.g.,* the works about generalization error bounds for learning problems (Germain et al., 2009; Welling & Teh, 2011; Parrado-Hernández et al., 2012; Dwork et al., 2015a;b; Blundell et al., 2015; Alquier et al., 2016; Thiemann et al., 2017; Letarte et al., 2019; Rivasplata et al., 2020; Pérez-Ortiz et al., 2021), especially the parametrization of the PAC-Bayesian bound with Gaussian distribution (Seeger, 2002; Langford & Caruana, 2002; Langford & Shawe-Taylor, 2003; Dziugaite & Roy, 2018; Jin et al., 2022). Specifically, Langford & Shawe-Taylor (2003); Germain et al. (2009); Parrado-Hernández et al. (2012) demonstrate that one can obtain tight PAC-Bayesian generalization bounds through multiplying the weight vector norm with a constant – this is different from the idea of expanding weight volume, as weight volume is a property of the weight covariance matrix, while weight vector norm is a property directly over weights.

Alquier et al. (2016) conducts a general study of the properties of fast Gibbs posterior approximations under PAC-Bayes. Letarte et al. (2019) uses the PAC-Bayesian theory to give a complete analysis of multilayer neural networks with binary activation. McAllester (2013) defines a natural notion of an uncorrelated posterior and provides a PAC-Bayesian bound for dropout training, while we consider the correlated posterior. It is interesting that Alquier et al. (2016) also optimizes PAC-Bayesian bound through a Gaussian posterior with a full covariance matrix. We use a similar assumption (correlated Gaussian posterior), but the main differences are that we develop it under another PAC-Bayesian branch (Dziugaite & Roy, 2017; Neyshabur et al., 2017; 2018a) and apply it on DNNs with connection to dropout noise.

### 6.4 Weight expansion and flatness

The flatness of the loss curve is thought to be related to the generalization performance of machine learning models (Hochreiter & Schmidhuber, 1994; 1997), especially DNNs (Dinh et al., 2017; Chaudhari et al., 2017; Yao et al., 2018; Liang et al., 2019). For example, in pioneering works (Hochreiter & Schmidhuber, 1994; 1997), the minimum description length (MDL) principle (Rissanen, 1978) was used to show that the flatness of the minimum is a good generalization measure to look at. Keskar et al. (2017) studies the reason for the generalization drop in large-batch networks and shows that large-batch algorithms tend to converge to sharp minima, resulting in worse generalization. Jiang et al. (2019) conducts a large-scale empirical analysis and discovers that flatness-based measures have a stronger correlation with generalization than weight norms and (margin and optimization)-based measures. Foret et al. (2021) proposes Sharpness-Aware Minimization (SAM) to find parameters that lie in neighborhoods with uniformly low loss, as they think gradient descent can be performed more efficiently with these parameters.

Of most relevance is the work by Dziugaite & Roy (2017); Neyshabur et al. (2017; 2018a), which relates a new concept of expected flatness over random weights to the PAC-Bayesian framework. By relaxing their i.i.d. assumption of random weights, our proposed concept of weight volume generalizes the expected flatness (that considers solely the *variance* of random weights) with weight correlation taken into account (that considers the weight *covariance*). Specifically, weight volume measures the expected flatness using the determinant of normalized covariance matrix of random weights. That is, a larger weight volume means random weights lie in a flatter local minimum (obeying a more spherical distribution), which can lead to better generalization. In particular, assuming a quadratic approximation of the loss holds, say Laplace approximation, weight volume approximately represents the inverse of the product of *all* eigenvalues of the Hessian (without normalization). That is, $\text{vol}(\mathbf{w}) \approx \prod_i \frac{1}{\lambda_i}$, where $\lambda_i$ is the $i$-th eigenvalue of the Hessian matrix $H$. Not only does this confirm the connection between weight volume and sharp/flat minima (i.e., large/small eigenvalues) of the Hessian, but also provides another, perhaps more comprehensive, view of flatness with respect to the *whole spectra* of the Hessian.

## 7 Conclusion and future work

This paper introduces 'weight expansion', an intriguing phenomenon that appears whenever dropout is applied during training. From a perspective of covariance, weight volume can be seen as a measure of flatness and weight expansion can be thought as random weights lying in a more spherical (flatter) local minimum. We have provided both theoretical and empirical arguments, showing that weight expansion can be one of the key indicators of the reduced generalization error, and that dropout is effective because it is (an inexpensive) method to obtain such weight expansion.

**Future work.** The following few directions may be worthy of future exploration and research. First, at the moment, we consider weights as a vector, without taking into consideration the structural information that the weights can be closely coupled according to the structures of the layers. It is interesting to know whether or not weight volume may capture some structural information of network (*e.g.,* convolutional layers). Second, as activation functions have been argued to have significant impact on not only the trained models but also the training process, it is interesting to study how weight volume is related to activation functions (*e.g.,* ReLU, tanh). Last but not least, it could be a significant topic to study whether the weight volume – when combined with the PAC-Bayes and adversarial samples (Farnia et al., 2018) – can provide any guidance to the adversarial robustness.

**Acknowledgement.** This project has received funding from the European Union's Horizon 2020 research and innovation programme under grant agreement No 956123, and from UK Dstl under project SOLITUDE. GJ and XH are also partially supported by the UK EPSRC under projects [EP/R026173/1, EP/T026995/1]. This work is also supported by the CAS Project for Young Scientists in Basic Research, Grant No.YSBR-040.

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

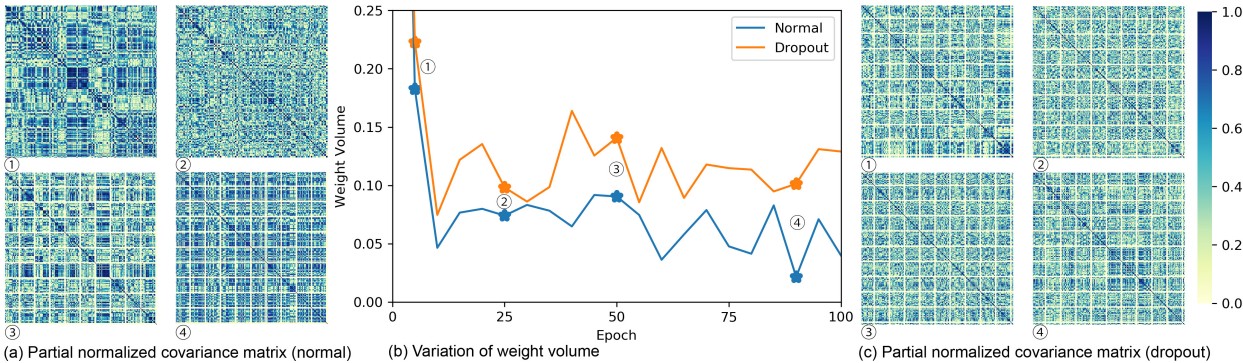

Figure 5: We have trained two VGG16 networks (with/without dropout) on CIFAR-10. **(b)** shows the changes in weight volume along with epochs. There are four sampling points (①②③④) to track the absolute normalized covariance matrix of normal gradient updates **(a)** and dropout gradient updates **(c)**.

Table 3: Network Structure for Figure 1.

| | | | | | | | |
|---|---|---|---|---|---|---|---|
| Normal CNN | Conv-64-(3,3) | Conv-128-(3,3) | MaxPool-(7,7) | Dense-128 | | Dense-2 | Softmax |
| Dropout CNN | Conv-64-(3,3) | Conv-128-(3,3) | MaxPool-(7,7) | Dense-128 | Dropout-0.3 | Dense-2 | Softmax |

## A    Details of Figure 1

As Table 3 shows, we use the above two networks to train MNIST with ReLU activation function, 0.01 learning rate, 128 minibatch size, stochastic gradient descent (SGD), 150 epochs, with and without dropout. Normal CNN and dropout CNN contain multiple convolutional layers with 64 and 128 channels, following with 128 and 2 hidden units. We visualize the features of Dense-2 layer (layer with 2 units) in Figure 1.

## B    Weight volume and weight orthogonality

For clarification, weight volume is not the orthogonality of weight matrices, but the "normalized generalized variance" of weight matrices. It is a statistical property of weight matrices when treated as *random variables*, which is different from the orthogonality technique in Saxe et al. (2013); Mishkin & Matas (2015); Bansal et al. (2018) that treat weight matrices as *deterministic variables*.

Specifically, by treating the elements in weight matrices as random variables (to fit the PAC-Bayes framework), the weight volume is defined as the determinant of the "covariance matrix" of weight matrices (see Definition 3.1). The optimization of the determinant of the covariance will not necessarily lead to orthogonality of weight matrices. It appears that weight volume considers the statistical correlation between any two weights (treated as random variables), whereas weight orthogonality considers the linear space correlation between any two columns in weight matrices.

*E.g.,* multivariate random variable $\mathbf{w}$ is made up of two terms, say $\mathbf{w} = \text{vec}(W) + \mathbf{U}_W$, where $\text{vec}(W)$ is the vectorization of weight matrix and $\mathbf{U}_W$ is the 0 mean multivariate random variable. Weight volume relies on the covariance matrix of $\mathbf{U}_W$, while weight orthogonality depends on $W$.

## C    Supplementary for Figure 2

Figure 5 also shows that the correlation among gradient updates in the normal network (Figure 5a) is larger than the correlation in dropout network (Figure 5c) and large correlation of gradient updates can lead to small weight volume (large correlation of weights), while small correlation of gradient updates can lead to large weight volume (small correlation of weights).

# D  Detailed derivation of PAC-Bayes

McAllester (1999); Dziugaite & Roy (2017) consider a training set $S$ with $m \in \mathbb{N}$ samples drawn from a distribution $D$. Given the prior and posterior distributions $P$ and $Q$ on the weights $\mathbf{w}$ respectively, for any $\delta > 0$, with probability $1 - \delta$ over the draw of training set, we have

$$\mathbb{E}_{\mathbf{w} \sim Q}[\mathcal{L}_D(f_{\mathbf{w}})] \leq \mathbb{E}_{\mathbf{w} \sim Q}[\mathcal{L}_S(f_{\mathbf{w}})] + \sqrt{\frac{D_{\mathrm{KL}}(Q||P) + \ln \frac{m}{\delta}}{2(m-1)}}. \tag{12}$$

Equation 4: Let $P = \mathcal{N}(\mu_P, \Sigma_P)$ and $Q = \mathcal{N}(\mu_Q, \Sigma_Q)$, we can get

$$
\begin{aligned}
D_{\mathrm{KL}}(\mathcal{N}(\mu_Q, \Sigma_Q)||\mathcal{N}(\mu_P, \Sigma_P)) &= \int [\ln(Q(x)) - \ln(P(x))]Q(x)dx \\
&= \int [\frac{1}{2}\ln\frac{\det\Sigma_P}{\det\Sigma_Q} - \frac{1}{2}(x-\mu_Q)^\intercal \Sigma_Q^{-1}(x-\mu_Q) + \frac{1}{2}(x-\mu_P)^\intercal \Sigma_P^{-1}(x-\mu_P)]Q(x)dx \\
&= \frac{1}{2}\ln\frac{\det\Sigma_P}{\det\Sigma_Q} - \frac{1}{2}\mathrm{tr}\{\mathbb{E}[(x-\mu_Q)(x-\mu_Q)^\intercal]\Sigma_Q^{-1}\} + \frac{1}{2}\mathrm{tr}\{\mathbb{E}[(x-\mu_P)^\intercal \Sigma_P^{-1}(x-\mu_P)]\} \\
&= \frac{1}{2}\ln\frac{\det\Sigma_P}{\det\Sigma_Q} - \frac{1}{2}\mathrm{tr}(I_k) + \frac{1}{2}(\mu_Q - \mu_P)^\intercal \Sigma_P^{-1}(\mu_Q - \mu_P) + \frac{1}{2}\mathrm{tr}(\Sigma_P^{-1}\Sigma_Q) \\
&= \frac{1}{2}\Big[\mathrm{tr}(\Sigma_P^{-1}\Sigma_Q) + (\mu_Q - \mu_P)^\intercal \Sigma_P^{-1}(\mu_Q - \mu_P) - k + \ln\frac{\det(\Sigma_P)}{\det(\Sigma_Q)}\Big].
\end{aligned} \tag{13}
$$

Equation 5: Neyshabur et al. (2017); Jiang et al. (2019) instantiate the prior $P$ to be a $\mathrm{vec}(W^0)$ mean, $\sigma^2$ variance Gaussian distribution and the posterior $Q$ to be a $\mathrm{vec}(W^F)$ mean spherical Gaussian with variance $\sigma^2$ in each direction. Relax their assumption, we assume $Q$ is a non-spherical Gaussian (**the off-diagonal correlations for same layer are not** 0**, those for different layers are** 0**),** then we can get

$$
\begin{aligned}
D_{\mathrm{KL}}(Q||P) &= \frac{||W^F - W^0||_F^2}{2\sigma^2} + \frac{1}{2}\ln\frac{\det(\Sigma_P)}{\det(\Sigma_Q)} \\
&= \sum_l \frac{||W_l^F - W_l^0||_F^2}{2\sigma^2} + \frac{1}{2}\ln\prod_l \frac{\det(\Sigma_{l,P})}{\det(\Sigma_{l,Q})} \\
&= \frac{1}{2}\sum_l \Big(\frac{||W_l^F - W_l^0||_F^2}{\sigma^2} + \ln\frac{\det(\Sigma_{l,P})}{\det(\Sigma_{l,Q})}\Big) \\
&= \frac{1}{2}\sum_l \Big(\frac{||W_l^F - W_l^0||_F^2}{\sigma^2} + \ln\frac{\prod_i [\Sigma_{l,P}]_{ii}}{\det(\Sigma_{l,Q})}\Big) \\
&= \frac{1}{2}\sum_l \Big(\frac{||W_l^F - W_l^0||_F^2}{\sigma^2} + \ln\frac{\prod_i [\Sigma_{l,Q}]_{ii}}{\det(\Sigma_{l,Q})}\Big) \\
&= \frac{1}{2}\sum_l \Big(\frac{||W_l^F - W_l^0||_F^2}{\sigma^2} + \ln\frac{1}{\mathrm{vol}(\mathbf{w}_l)}\Big).
\end{aligned} \tag{14}
$$

# E   Proof of Lemma 3.2

**Proof E.1** *Let $\Delta_{ij}$, $\Delta_{i'j'}$ be the gradient updates for $\mathbf{w}_{ij}$, $\mathbf{w}_{i'j'}$ of some layer in a normal network, $i \neq i'$, $\tilde{\Delta}_{ij}$ and $\tilde{\Delta}_{i'j'}$ be the gradient updates for $\mathbf{w}_{ij}$, $\mathbf{w}_{i'j'}$ in the same setting dropout network. Let $\text{vec}(\Delta) \sim \mathcal{M}_\Delta(\mu_\Delta, \Sigma_\Delta)$ and $\text{vec}(\tilde{\Delta}) = (\mathbf{1} \otimes \tilde{\eta}) \odot \text{vec}(\tilde{\Delta}')$, $\text{vec}(\tilde{\Delta}') \sim \mathcal{M}_\Delta(\mu_\Delta, \Sigma_\Delta)$. **Note that, as we aim to study the impact of $\tilde{\eta}$ on the correlation among gradient updates, we assume $\text{vec}(\Delta)$ and $\text{vec}(\tilde{\Delta}')$ are i.i.d.**. As dropout noises for different nodes (units) are independent, we have*

$$
\begin{aligned}
\text{Cov}(\tilde{\Delta}_{ij}, \tilde{\Delta}_{i'j'}) &= \mathbb{E}_{\tilde{\Delta}}\left[(\tilde{\Delta}_{ij} - \mu_{\tilde{\Delta}_{ij}})(\tilde{\Delta}_{i'j'} - \mu_{\tilde{\Delta}_{i'j'}})\right] \\
&= (1-q)^2 \mathbb{E}_{\tilde{\Delta}'}\left[(\frac{\tilde{\Delta}'_{ij}}{1-q} - \mu_{\Delta_{ij}})(\frac{\tilde{\Delta}'_{i'j'}}{1-q} - \mu_{\Delta_{i'j'}})\right] + (1-q)q \\
&\quad \mathbb{E}_{\tilde{\Delta}'}\left[(\frac{\tilde{\Delta}'_{i'j'}}{1-q} - \mu_{\Delta_{i'j'}})(-\mu_{\Delta_{ij}})\right] + (1-q)q\mathbb{E}_{\tilde{\Delta}'}\left[(\frac{\tilde{\Delta}'_{ij}}{1-q} - \mu_{\Delta_{ij}})(-\mu_{\Delta_{i'j'}})\right] \\
&\quad + q^2\left[(-\mu_{\Delta_{i'j'}})(-\mu_{\Delta_{ij}})\right] \\
&= \mathbb{E}_{\tilde{\Delta}'}\left[(\tilde{\Delta}'_{ij} - \mu_{\Delta_{ij}})(\tilde{\Delta}'_{i'j'} - \mu_{\Delta_{i'j'}})\right] \\
&= \mathbb{E}_\Delta\left[(\Delta_{ij} - \mu_{\Delta_{ij}})(\Delta_{i'j'} - \mu_{\Delta_{i'j'}})\right] \\
&= \text{Cov}(\Delta_{ij}, \Delta_{i'j'}),
\end{aligned}
\tag{15}
$$

*and*

$$
\begin{aligned}
\text{Var}(\tilde{\Delta}_{ij}) &= \mathbb{E}_{\tilde{\Delta}}\left[(\tilde{\Delta}_{ij} - \mu_{\tilde{\Delta}_{ij}})^2\right] \\
&= (1-q)\mathbb{E}_{\tilde{\Delta}'}\left[(\frac{\tilde{\Delta}'_{ij}}{1-q} - \mu_{\Delta_{ij}})^2\right] + q(-\mu_{\Delta_{ij}})^2 \\
&= \frac{\mathbb{E}_{\tilde{\Delta}'}\left[(\tilde{\Delta}'_{ij} - \mu_{\Delta_{ij}})^2\right]}{1-q} + \frac{q\mu^2_{\Delta_{ij}}}{1-q} \\
&= \frac{\mathbb{E}_\Delta\left[(\Delta_{ij} - \mu_{\Delta_{ij}})^2\right]}{1-q} + \frac{q\mu^2_{\Delta_{ij}}}{1-q} \\
&= \frac{\text{Var}(\Delta_{ij})}{1-q} + \frac{q\mu^2_{\Delta_{ij}}}{1-q}.
\end{aligned}
\tag{16}
$$

*Similarly, $\text{Var}(\tilde{\Delta}_{i'j'}) = \frac{\text{Var}(\Delta_{i'j'})}{1-q} + \frac{q\mu^2_{\Delta_{i'j'}}}{1-q}$. Thus, we have*

$$
\begin{aligned}
\left|\rho_{\tilde{\Delta}_{ij}, \tilde{\Delta}_{i'j'}}\right| &= \frac{\left|\text{Cov}(\tilde{\Delta}_{ij}, \tilde{\Delta}_{i'j'})\right|}{\sqrt{\text{Var}(\tilde{\Delta}_{ij})\text{Var}(\tilde{\Delta}_{i'j'})}} \\
&= \frac{\left|\text{Cov}(\Delta_{ij}, \Delta_{i'j'})\right|}{\sqrt{\left(\frac{\text{Var}(\Delta_{ij})}{1-q} + \frac{q\mu^2_{\Delta_{ij}}}{1-q}\right)\left(\frac{\text{Var}(\Delta_{i'j'})}{1-q} + \frac{q\mu^2_{\Delta_{i'j'}}}{1-q}\right)}} \\
&\leq \frac{(1-q)\left|\text{Cov}(\Delta_{ij}, \Delta_{i'j'})\right|}{\sqrt{\text{Var}(\Delta_{ij})\text{Var}(\Delta_{i'j'})}} \\
&= (1-q)\left|\rho_{\Delta_{ij}, \Delta_{i'j'}}\right| \\
&\leq \left|\rho_{\Delta_{ij}, \Delta_{i'j'}}\right|.
\end{aligned}
\tag{17}
$$

# F   Proof of Lemma 3.3

**Proof F.1** *As $\mathbf{w} \sim \mathcal{N}_{\mathbf{w}}$, we have*

$$
\begin{aligned}
\mathrm{Cov}(\mathbf{w}_{ij}, \tilde{\Delta}_{i'j'}) &= \mathbb{E}_{\mathbf{w}, \tilde{\Delta}}\big[(\mathbf{w}_{ij} - \mu_{\mathbf{w}_{ij}})(\tilde{\Delta}_{i'j'} - \mu_{\tilde{\Delta}_{i'j'}})\big] \\
&= (1-q)\mathbb{E}_{\mathbf{w}, \tilde{\Delta}'}\big[(\mathbf{w}_{ij} - \mu_{\mathbf{w}_{ij}})(\frac{\tilde{\Delta}'_{i'j'}}{1-q} - \mu_{\Delta_{i'j'}})\big] + q\mathbb{E}_{\mathbf{w}}\big[(\mathbf{w}_{ij} - \mu_{\mathbf{w}_{ij}})(-\mu_{\Delta_{i'j'}})\big] \\
&= \mathbb{E}_{\mathbf{w}, \tilde{\Delta}'}\big[(\mathbf{w}_{ij} - \mu_{\mathbf{w}_{ij}})(\tilde{\Delta}'_{i'j'} - \mu_{\Delta_{i'j'}})\big] \\
&= \mathbb{E}_{\mathbf{w}, \Delta}\big[(\mathbf{w}_{ij} - \mu_{\mathbf{w}_{ij}})(\Delta_{i'j'} - \mu_{\Delta_{i'j'}})\big] \\
&= \mathrm{Cov}(\mathbf{w}_{ij}, \Delta_{i'j'}),
\end{aligned}
\tag{18}
$$

$\mathrm{Cov}(\mathbf{w}_{ij}, \tilde{\Delta}_{ij})$, $\mathrm{Cov}(\mathbf{w}_{i'j'}, \tilde{\Delta}_{i'j'})$ and $\mathrm{Cov}(\mathbf{w}_{i'j'}, \tilde{\Delta}_{ij})$ are similar. From Equations 15, 16, we can get

$$
\begin{aligned}
&\big|\rho_{\mathbf{w}_{ij}+\tilde{\Delta}_{ij}, \mathbf{w}_{i'j'}+\tilde{\Delta}_{i'j'}}\big| \\
&= \frac{\big|\mathrm{Cov}(\mathbf{w}_{ij}+\tilde{\Delta}_{ij}, \mathbf{w}_{i'j'}+\tilde{\Delta}_{i'j'})\big|}{\sqrt{\mathrm{Var}(\mathbf{w}_{ij}+\tilde{\Delta}_{ij})\mathrm{Var}(\mathbf{w}_{i'j'}+\tilde{\Delta}_{i'j'})}} \\
&= \frac{\big|\mathrm{Cov}(\mathbf{w}_{ij}, \mathbf{w}_{i'j'}) + \mathrm{Cov}(\tilde{\Delta}_{ij}, \tilde{\Delta}_{i'j'}) + \mathrm{Cov}(\mathbf{w}_{ij}, \tilde{\Delta}_{i'j'}) + \mathrm{Cov}(\mathbf{w}_{i'j'}, \tilde{\Delta}_{ij})\big|}{\sqrt{\big(\mathrm{Var}(\mathbf{w}_{ij}) + \mathrm{Var}(\tilde{\Delta}_{ij}) + 2\mathrm{Cov}(\mathbf{w}_{ij}, \tilde{\Delta}_{ij})\big)\big(\mathrm{Var}(\mathbf{w}_{i'j'}) + \mathrm{Var}(\tilde{\Delta}_{i'j'}) + 2\mathrm{Cov}(\mathbf{w}_{i'j'}, \tilde{\Delta}_{i'j'})\big)}} \\
&= \frac{\big|\mathrm{Cov}(\mathbf{w}_{ij}, \mathbf{w}_{i'j'}) + \mathrm{Cov}(\Delta_{ij}, \Delta_{i'j'}) + \mathrm{Cov}(\mathbf{w}_{ij}, \Delta_{i'j'}) + \mathrm{Cov}(\mathbf{w}_{i'j'}, \Delta_{ij})\big|}{\sqrt{\big(\mathrm{Var}(\mathbf{w}_{ij}) + \mathrm{Var}(\tilde{\Delta}_{ij}) + 2\mathrm{Cov}(\mathbf{w}_{ij}, \Delta_{ij})\big)\big(\mathrm{Var}(\mathbf{w}_{i'j'}) + \mathrm{Var}(\tilde{\Delta}_{i'j'}) + 2\mathrm{Cov}(\mathbf{w}_{i'j'}, \Delta_{i'j'})\big)}} \\
&= \frac{\big|\mathrm{Cov}(\mathbf{w}_{ij}, \mathbf{w}_{i'j'}) + \mathrm{Cov}(\Delta_{ij}, \Delta_{i'j'}) + \mathrm{Cov}(\mathbf{w}_{ij}, \Delta_{i'j'}) + \mathrm{Cov}(\mathbf{w}_{i'j'}, \Delta_{ij})\big|}{\sqrt{\big(\mathrm{Var}(\mathbf{w}_{ij}) + \frac{\mathrm{Var}(\Delta_{ij})}{1-q} + \frac{q\mu^2_{\Delta_{ij}}}{1-q} + 2\mathrm{Cov}(\mathbf{w}_{ij}, \Delta_{ij})\big)}} \\
&\quad \frac{1}{\sqrt{\big(\mathrm{Var}(\mathbf{w}_{i'j'}) + \frac{\mathrm{Var}(\Delta_{i'j'})}{1-q} + \frac{q\mu^2_{\Delta_{i'j'}}}{1-q} + 2\mathrm{Cov}(\mathbf{w}_{i'j'}, \Delta_{i'j'})\big)}} \\
&\leq \frac{\big|\mathrm{Cov}(\mathbf{w}_{ij}+\Delta_{ij}, \mathbf{w}_{i'j'}+\Delta_{i'j'})\big|}{\sqrt{\big(\mathrm{Var}(\mathbf{w}_{ij}+\Delta_{ij}) + \frac{q\mathrm{Var}(\Delta_{ij})}{1-q}\big)\big(\mathrm{Var}(\mathbf{w}_{i'j'}+\Delta_{i'j'}) + \frac{q\mathrm{Var}(\Delta_{i'j'})}{1-q}\big)}} \\
&\leq \frac{\big|\mathrm{Cov}(\mathbf{w}_{ij}+\Delta_{ij}, \mathbf{w}_{i'j'}+\Delta_{i'j'})\big|}{\sqrt{\mathrm{Var}(\mathbf{w}_{ij}+\Delta_{ij})\mathrm{Var}(\mathbf{w}_{i'j'}+\Delta_{i'j'})}} \\
&= \big|\rho_{\mathbf{w}_{ij}+\Delta_{ij}, \mathbf{w}_{i'j'}+\Delta_{i'j'}}\big|.
\end{aligned}
$$

# G   Weight volume and correlation

Since the definition of weight volume in Definition 3.1, the weight volume equals the determinant of the weight correlation matrix.

It is difficult to theoretically argue an 'exact' relationship (*e.g.,* perfect positive or negative) between weight volume and correlation of weights, but they have a rough negative relationship. Take a simple example, let $n$-dimensional weight correlation matrix $C$ have the same correlation $\rho$, it is easy to infer that (absolute) correlation is negative related with weight volume, as

$$
\det(C) = (1-\rho)^{n-1}\big(1 + (n-1)\rho\big),
\tag{19}
$$

where $\frac{d \det(C)}{d\rho} = (n - n^2)\rho(1-\rho)^{n-2}$ and $n \geq 2$. Also, we demonstrate the relationship between weight volume and average (absolute) correlation with 10000 samples for a 3-dimensional correlation matrix (Figure 6a) and a 4-dimensional correlation matrix (Figure 6b) respectively. Although weight volume is not monotonically increasing as (absolute) correlation decreases, we can find a rough negative relationship.

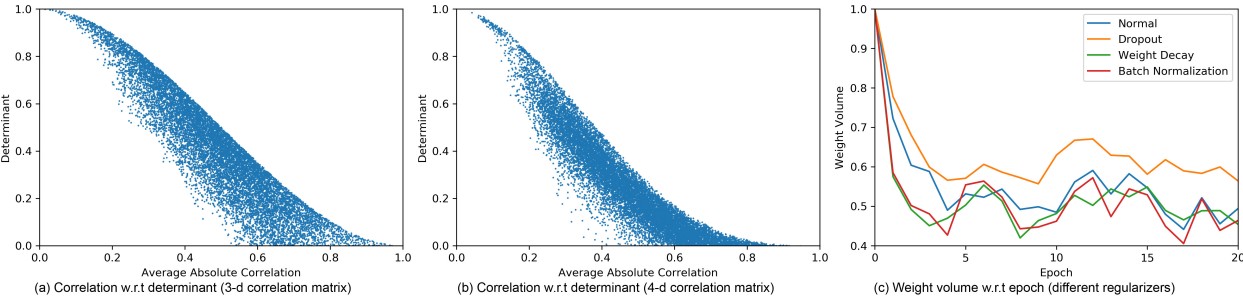

Figure 6: **(a)** We sample 10000 3-dimensional correlation matrices and show their determinant and average absolute correlation. **(b)** We sample 10000 4-dimensional correlation matrices and show their determinant and average absolute correlation. **(c)** We train four small NNs (64-32-16-10 with normal, dropout, weight decay, and batch normalization, respectively) on MNIST. The figure shows the changes in weight volume along with epochs for these four networks.

## H  Other regularizers with weight volume

As dropout can generate independent noises but other regularizers, like weight decay and batch normalization, cannot. Thus, dropout can introduce randomness into the network, while weight decay and batch normalization cannot (these two regularizers improve generalization not through weight volume, through other keys). We have described in detail why this dropout noise can expand weight volume and why weight volume is correlated with generalization, theoretically (Section 3) and empirically (Section 5, Figures 2, 5).

Also, in Figure 6c, we can see that dropout expands weight volume compared with normal network, while weight decay and batch normalization have hardly any impact on weight volume.

## I  Estimate weight volume

The approximation of $\mathbb{E}_s[H_l]$ is two-fold: (1) Only the diagonal blocks of the weight matrix are captured, so that the correlation between blocks is decoupled; (2) Assume $\mathcal{A}_{l-1}$ and $\mathcal{H}_l$ are independent (Botev et al., 2017; Ritter et al., 2018). Then, we have

$$\mathbb{E}_s[H_l] = \mathbb{E}_s[\mathcal{A}_{l-1} \otimes \mathcal{H}_l] \approx \mathbb{E}_s[\mathcal{A}_{l-1}] \otimes \mathbb{E}_s[\mathcal{H}_l]. \tag{20}$$

Finally, $\Sigma_l$ can be approximated as follows:

$$\Sigma_l = (\mathbb{E}_s[H_l])^{-1} \approx (\mathbb{E}_s[\mathcal{A}_{l-1}])^{-1} \otimes (\mathbb{E}_s[\mathcal{H}_l])^{-1}. \tag{21}$$

Therefore, by separately computing the matrix inverse of $\mathbb{E}_s[\mathcal{A}_{l-1}]$ and $\mathbb{E}_s[\mathcal{H}_l]$, whose sizes are significantly smaller, we can estimate $\Sigma_l \in \mathbb{R}^{N_{l-1}N_l \times N_{l-1}N_l}$ efficiently.

Further, the weight volume can be approximated by using

$$\det(\Sigma_l) \approx \det\left((\mathbb{E}_s[\mathcal{A}_{l-1}])^{-1}\right)^{N_l} \cdot \det\left((\mathbb{E}_s[\mathcal{H}_l])^{-1}\right)^{N_{l-1}} \tag{22}$$

and

$$\prod_i [\Sigma_l]_{ii} \approx \left(\prod_c \left[(\mathbb{E}_s[\mathcal{A}_{l-1}])^{-1}\right]_{cc}\right)^{N_l} \cdot \left(\prod_d \left[(\mathbb{E}_s[\mathcal{H}_l])^{-1}\right]_{dd}\right)^{N_{l-1}}. \tag{23}$$

## J  Supplementary for Section 5.1

As Appendix 4 shows, on the ImageNet-32, dropout still leads to persistent, and significant, increase on the weight volume, across all three models. At the same time, the generalization error is reduced.

Table 4: Weight Volume on VGG Networks (ImageNet-32, weight decay = 0).

| Method | VGG11 | VGG11(dropout) | VGG16 | VGG16(dropout) | VGG19 | VGG19(dropout) |
|---|---|---|---|---|---|---|
| Sampling | 0.015±0.01 | **0.089±0.01** | 0.016±0.01 | **0.097±0.01** | 0.017±0.01 | **0.086±0.01** |
| Laplace | 0.0198 | **0.0985** | 0.0194 | **0.0942** | 0.0214 | **0.0977** |
| Generalization Gap | 7.0388 | **2.0501** | 6.3922 | **1.2430** | 7.5187 | **0.7910** |

Table 5: VGG11 Structure

| Conv-64 | Maxpooling | Conv-128 | Maxpooling | Conv-256 | Dropout(0.3) | Conv-256 | Maxpooling | Conv-512 | Dropout(0.3) |
|---|---|---|---|---|---|---|---|---|---|
| Conv-512 | Maxpooling | Conv-512 | Dropout(0.3) | Conv-512 | Maxpooling | Dropout(0.1) | Dense-512 | Dropout(0.1) | Dense-(num classes) |

Table 6: VGG16 Structure

| Conv-64 | Dropout(0.3) | Conv-64 | Maxpooling | Conv-128 | Dropout(0.3) | Conv-128 | Maxpooling | Conv-256 | Dropout(0.3) |
|---|---|---|---|---|---|---|---|---|---|
| Conv-256 | Dropout(0.3) | Conv-256 | Maxpooling | Conv-512 | Dropout(0.3) | Conv-512 | Dropout(0.3) | Conv-512 | Maxpooling |
| Conv-512 | Dropout(0.3) | Conv-512 | Dropout(0.3) | Conv-512 | Maxpooling | Dropout(0.1) | Dense-512 | Dropout(0.1) | Dense-(num classes) |

Table 7: VGG19 Structure

| Conv-64 | Dropout(0.3) | Conv-64 | Maxpooling | Conv-128 | Dropout(0.3) | Conv-128 | Maxpooling | Conv-256 | Dropout(0.3) |
|---|---|---|---|---|---|---|---|---|---|
| Conv-256 | Dropout(0.3) | Conv-256 | Dropout(0.3) | Conv-256 | Maxpooling | Conv-512 | Dropout(0.3) | Conv-512 | Dropout(0.3) |
| Conv-512 | Dropout(0.3) | Conv-512 | Maxpooling | Conv-512 | Dropout(0.3) | Conv-512 | Dropout(0.3) | Conv-512 | Dropout(0.3) |
| Conv-512 | Maxpooling | Dropout(0.1) | Dense-512 | Dropout(0.1) | Dense-(num classes) | | | | |

Table 8: Weight Volume on VGG Networks (ImageNet-32, weight decay = 0).

| Method | VGG11 | VGG11(dropout) | VGG16 | VGG16(dropout) |
|---|---|---|---|---|
| Sampling | 0.015±0.01 | **0.089±0.01** | 0.016±0.01 | **0.097±0.01** |
| Laplace | 0.0198 | **0.0985** | 0.0194 | **0.0942** |
| Train Loss | 0.0093 | 1.1571 | 0.0085 | 1.6577 |
| Test Loss | 7.0481 | 3.2072 | 6.4007 | 2.9007 |
| Loss Gap | 7.0388 | **2.0501** | 6.3922 | **1.2430** |
| Top-1 Train Acc | 0.9953 | 0.7088 | 0.9953 | 0.5949 |
| Top-1 Test Acc | 0.2738 | 0.3694 | 0.3116 | 0.3908 |
| Top-1 Acc Gap | 0.7215 | **0.3394** | 0.6837 | **0.2041** |
| Top-5 Train Acc | 0.9999 | 0.8023 | 0.9999 | 0.8248 |
| Top-5 Test Acc | 0.5014 | 0.6170 | 0.5502 | 0.6444 |
| Top-5 Acc Gap | 0.4985 | **0.1853** | 0.4497 | **0.1804** |

Table 9: Weight Volume on VGG Networks (ImageNet-32, weight decay = 0.0005).

| Method | VGG11 | VGG11(dropout) | VGG16 | VGG16(dropout) |
|---|---|---|---|---|
| Sampling | 0.021±0.01 | **0.097±0.01** | 0.019±0.01 | **0.091±0.01** |
| Laplace | 0.0268 | **0.0877** | 0.0284 | **0.0891** |
| Train Loss | 0.7344 | 2.3641 | 0.6708 | 2.6208 |
| Test Loss | 4.8745 | 3.2244 | 4.6388 | 3.0396 |
| Loss Gap | 4.1410 | **0.8603** | 3.9680 | **0.4188** |
| Top-1 Train Acc | 0.9926 | 0.5312 | 0.9932 | 0.4849 |
| Top-1 Test Acc | 0.3265 | 0.4033 | 0.3732 | 0.4266 |
| Top-1 Acc Gap | 0.6661 | **0.1279** | 0.6200 | **0.0583** |
| Top-5 Train Acc | 0.9999 | 0.7839 | 0.9999 | 0.7419 |
| Top-5 Test Acc | 0.5691 | 0.6531 | 0.6145 | 0.6777 |
| Top-5 Acc Gap | 0.4308 | **0.1308** | 0.3854 | **0.0642** |

Table 10: Weight volume on VGG networks (weight decay = 0).

| | Method | VGG11 | VGG11(dropout) | VGG16 | VGG16(dropout) |
|---|---|---|---|---|---|
| CIFAR-10 | Sampling | 0.06±0.02 | **0.13±0.02** | 0.05±0.02 | **0.12±0.02** |
| | Laplace | 0.0568 | **0.1523** | 0.0475 | **0.1397** |
| | Train Loss | 0.0083 | 0.0806 | 0.0203 | 0.3002 |
| | Test Loss | 0.8113 | 0.6021 | 0.8901 | 0.5998 |
| | Loss Gap | 0.8030 | **0.5215** | 0.8698 | **0.2996** |
| | Train Acc | 0.9973 | 0.9698 | 0.9933 | 0.8813 |
| | Test Acc | 0.8563 | 0.8645 | 0.8435 | 0.8401 |
| | Acc Gap | 0.1410 | **0.1053** | 0.1498 | **0.0412** |
| CIFAR-100 | Sampling | 0.07±0.02 | **0.14±0.02** | 0.06±0.02 | **0.14±0.02** |
| | Laplace | 0.0537 | **0.1578** | 0.0409 | **0.1620** |
| | Train Loss | 0.0187 | 0.2608 | 0.0706 | 1.0324 |
| | Test Loss | 3.1395 | 2.4244 | 3.8247 | 2.3038 |
| | Loss Gap | 3.1208 | **2.1635** | 3.7541 | **1.2714** |
| | Train Acc | 0.9943 | 0.9274 | 0.9912 | 0.6398 |
| | Test Acc | 0.5764 | 0.5911 | 0.4867 | 0.5188 |
| | Acc Gap | 0.4179 | **0.3363** | 0.5045 | **0.1210** |

In addition, we show the VGG network structures in Tables 5, 6, 7, and also demonstrate the details (including training loss, training accuracy, test loss, test accuracy) of the VGG experiments on ImageNet-32 (Tables 8, 9) and CIFAR-10/100 (Tables 10, 11). All models for CIFAR-10/100 are trained for 140 epochs using SGD with momentum 0.9, batch size 128, weight decay 0 or $5 \times 10^{-4}$, and an initial learning rate of 0.1 that is

Table 11: Weight volume on VGG networks (weight decay = 0.0005).

| | Method | VGG11 | VGG11(dropout) | VGG16 | VGG16(dropout) |
|---|---|---|---|---|---|
| CIFAR-10 | Sampling | 0.05±0.02 | **0.13±0.02** | 0.05±0.02 | **0.13±0.02** |
| | Laplace | 0.0512 | **0.1427** | 0.0433 | **0.1346** |
| | Train Loss | 0.2166 | 0.2984 | 0.1739 | 0.2438 |
| | Test Loss | 0.6368 | 0.5997 | 0.5346 | 0.4788 |
| | Loss Gap | 0.4202 | **0.3013** | 0.3607 | **0.2350** |
| | Train Acc | 0.9940 | 0.9801 | 0.9958 | 0.9878 |
| | Test Acc | 0.9048 | 0.9118 | 0.9234 | 0.9340 |
| | Acc Gap | 0.0892 | **0.0683** | 0.0724 | **0.0538** |
| CIFAR-100 | Sampling | 0.05±0.02 | **0.15±0.02** | 0.06±0.02 | **0.15±0.02** |
| | Laplace | 0.0496 | **0.1469** | 0.0413 | **0.1659** |
| | Train Loss | 0.4389 | 0.7612 | 0.5403 | 0.8890 |
| | Test Loss | 2.4366 | 2.3672 | 2.3149 | 2.1516 |
| | Loss Gap | 1.9977 | **1.6060** | 1.7746 | **1.2626** |
| | Train Acc | 0.9976 | 0.9654 | 0.9889 | 0.9133 |
| | Test Acc | 0.6511 | 0.6765 | 0.6980 | 0.6970 |
| | Acc Gap | 0.3465 | **0.2889** | 0.2909 | **0.2163** |

divided by 2 after every 20 epochs. All models for ImageNet-32 are trained for 100 epochs using SGD with momentum 0.9, batch size 1024, weight decay 0 or $5 \times 10^{-4}$, and an initial learning rate of 0.1 that is divided by 2 for each 20 epochs, and learning rate of 0.0005 after 50th epoch.

All results in Tables 8, 9, 10, 11 also suggest that dropout can expand the weight volume and narrow generalization gap.

## K    Details of experiments in Section 5.2

### K.1   Experimental settings for Section 5.2

We consider both VGG-like models, with dropout rate={0.0, 0.15, 0.30}, parameterized over $\Omega_{\text{VGG}}$={batch size={64, 256}, initial learning rate={0.1, 0.03},[2] depth={11, 16, 19}, activation function={ReLU, tanh}, weight decay={0.0, 0.0005}, width={1.0, 0.5}[3] }, and AlexNet-like models, parameterized over $\Omega_{\text{Alex}}$, which has the same parameters except that depth={7, 9, 10}. Therefore, either VGG or AlexNet has $2*2*3*2*2*2*3 = 288$ configurations. Let $\Psi$ be the set of configurations.

Formally, we use a normalized **MI** to quantify the relation between random variables $V_{\mathbf{co}}$ and $V_{\mathbf{pa}}$ with $\mathbf{pa} \in \Omega \cup \{\mathbf{GG}\}$, where '$\mathbf{co}$' denotes complexity measure and '$\mathbf{GG}$' generalization gap. For example, $V_{\text{depth}}$ is the random variable for network depth.

Given a set of models, we have their associated hyper-parameter or generalization gap $\{\mathbf{pa}(f_{W,\psi})|\psi \in \Psi\}$, and their respective values of the complexity measure $\{\mathbf{co}(f_{W,\psi})|\psi \in \Psi\}$. Let $V_{\mathbf{co}}(\psi_i, \psi_j) \triangleq \text{sign}(\mathbf{co}(f_{W,\psi_i}) - \mathbf{co}(f_{W,\psi_j}))$, $V_{\mathbf{pa}}(\psi_i, \psi_j) \triangleq \text{sign}(\mathbf{pa}(f_{W,\psi_i}) - \mathbf{pa}(f_{W,\psi_j}))$. We can compute their joint probability $p(V_{\mathbf{co}}, V_{\mathbf{pa}})$ and marginal probabilities $p(V_{\mathbf{co}})$ and $p(V_{\mathbf{pa}})$. The **MI** between $V_{\mathbf{co}}$ and $V_{\mathbf{pa}}$ can be computed as

$$I(V_{\mathbf{co}}, V_{\mathbf{pa}}) = \sum_{V_{\mathbf{co}}} \sum_{V_{\mathbf{pa}}} p(V_{\mathbf{co}}, V_{\mathbf{pa}}) \log \frac{p(V_{\mathbf{co}}, V_{\mathbf{pa}})}{p(V_{\mathbf{co}})p(V_{\mathbf{pa}})}, \quad \hat{I}(V_{\mathbf{co}}, V_{\mathbf{pa}}) = \frac{I(V_{\mathbf{co}}, V_{\mathbf{pa}})}{H(V_{\mathbf{pa}})}, \tag{24}$$

where $I(V_{\mathbf{co}}, V_{\mathbf{pa}})$ can be further normalized as $\hat{I}(V_{\mathbf{co}}, V_{\mathbf{pa}})$ to enable a fair comparison between complexity measures, $H(\cdot)$ is the marginal entropy. For the generalization gap, we consider not only $\hat{I}(V_{\mathbf{co}}, V_{\mathbf{GG}})$, but also the **MI** with some of the hyper-parameters fixed, i.e., $\mathbb{E}_{\omega \subset \Omega}\big(\hat{I}(V_{\mathbf{co}}, V_{\mathbf{GG}}|\omega)\big)$, for the cases of $|\omega| = 0, 1, 2$. Note that, when $|\omega| = 0$, it is the same as $\hat{I}(V_{\mathbf{co}}, V_{\mathbf{GG}})$. Intuitively, $\mathbb{E}_{\omega \subset \Omega}\big(\hat{I}(V_{\mathbf{co}}, V_{\mathbf{GG}}|\omega)\big)$ with $|\omega| = 2$ is the expected value of $\hat{I}(V_{\mathbf{co}}, V_{\mathbf{GG}})$ when 2 hyper-parameters in $\Omega$ are fixed. Note that, as we aim to study the relationship between dropout and generalization, we only compute $\mathbb{E}_{\omega \subset \Omega}\big(\hat{I}(V_{\mathbf{co}}, V_{\mathbf{GG}}|\omega)\big)$ with $|\omega| = 0, 1, 2$ and $\hat{I}(V_{\mathbf{co}}, V_{\mathbf{dropout}})$ for all complexity measures.

---

[2]The learning rate decays 50% per 20 epochs.
[3]For VGG-like models, width = 1 means the model has 64, 128, 256, 512, 512 channels in its structure, while width = 0.5 means that the model has 32, 64, 128, 256, 256 channels in its structure.

Table 12: Mutual Information of Complexity Measures on CIFAR-100.

| Complexity | VGG | | | |
|---|---|---|---|---|
| Measure | Dropout | $\mathbf{GG}(|\omega|=2)$ | $\mathbf{GG}(|\omega|=1)$ | $\mathbf{GG}(|\omega|=0)$ |
| Frob Distance | 0.0128 | 0.0172 | 0.0068 | 0.0001 |
| Spectral Norm | 0.0109 | 0.0237 | 0.0141 | 0.0074 |
| Parameter Norm | 0.0122 | 0.0168 | 0.0086 | 0.0042 |
| Path Norm | 0.0370 | 0.0155 | 0.0025 | 0.0009 |
| Sharpness $\alpha'$ | 0.0333 | 0.0342 | 0.0164 | 0.0030 |
| PAC-Sharpness | 0.0390 | 0.0346 | 0.0166 | 0.0019 |
| **PAC-S(Laplace)** | 0.6225 | 0.1439 | 0.1256 | 0.1083 |
| **PAC-S(Sampling)** | 0.5364 | 0.1001 | 0.0852 | 0.0787 |

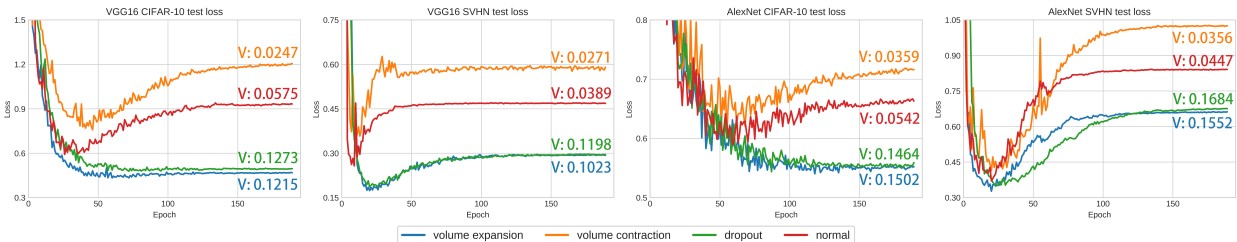

Figure 7: **Disentanglement noise v.s. dropout.** We train 2 VGG16 networks (**Left**) and 2 AlexNet (**Right**) on CIFAR-10 and SVHN respectively, with normal training, dropout training, disentanglement noise training (volume expansion) and stochastic noise training (volume contraction). For each dataset, we present their test losses in different plots.

### K.2 Complexity measures

In the following, we show the details of complexity measures.

**Frobenius (Frob) Distance and Parameter Norm** (Dziugaite & Roy, 2017; Nagarajan & Kolter, 2019; Neyshabur et al., 2018b):

$$\mathbf{co}_{\text{Frob}}(f_W) = \sum_l ||W_l^F - W_l^0||_F^2. \tag{25}$$

$$\mathbf{co}_{\text{Para}}(f_W) = \sum_l ||W_l^F||_F^2. \tag{26}$$

**Spectral Norm** (Yoshida & Miyato, 2017):

$$\mathbf{co}_{\text{Spec}}(f_W) = \sum_l ||W_l^F - W_l^0||_2^2. \tag{27}$$

**Path Norm:** Path-norm was introduced in (Neyshabur et al., 2015b) as an scale invariant complexity measure for generalization and is shown to be a useful geometry for optimization (Neyshabur et al., 2015a). To calculate path-norm, we square the parameters of the network, do a forward pass on an all-ones input and then take square root of sum of the network outputs. We define the following measures based on the path-norm:

$$\mathbf{co}_{\text{Path}}(f_W) = \sum_l f_{W^2}(1), \tag{28}$$

where $W^2 = W \circ W$ is the element-wise square operation on the parameters.

**Sharpness-based PAC-Bayes:**

$$\mathbf{co}_{\text{PAC}-\text{S}}(f_W) = \sum_l \frac{||W_l^F - W_l^0||_F^2}{2\sigma_l^2}, \tag{29}$$

where $\sigma$ is estimated by sharpness method. That is, let $\text{vec}(U') \sim \mathcal{N}(\mathbf{0}, \sigma'^2 I)$ be a sample from a zero mean Gaussian, where $\sigma$ is chosen to be the largest number such that $\max_{\sigma' \leq \sigma} |\mathcal{L}(f_{W+U'}) - \mathcal{L}(f_W)| \leq \epsilon$ (Keskar et al., 2016; Pitas et al., 2017; Neyshabur et al., 2017; Jiang et al., 2019).

**PAC-Bayes with Weight Volume:**

$$\mathbf{co}_{\text{PAC}-\text{S(Laplace/Sampling)}}(f_W) = \sum_l \left[ \frac{\|W_l^F - W_l^0\|_F^2}{2\sigma_l^2} + \ln \frac{1}{\text{vol}(\mathbf{w}_l)} \right], \tag{30}$$

where $\text{vol}(\mathbf{w}_l)$ is estimated by Laplace approximation (**PAC-S(Laplace)**) and sampling method (**PAC-S(Sampling)**).

### K.3 Supplementary for Section 5.2

As Table 12 shows, for all cases concerning the generalization error on CIFAR-100 (VGG), i.e., $|\omega| = 0,1,2$, PAC-S(Laplace) or PAC-S(Sampling) is still a clear winner and is also closely correlated with dropout.

## L Supplementary for Experiment 5.3

As Figure 7 shows, on the CIFAR-10 and SVHN, weight volume expansion improves generalization performance, similar to dropout. This is in contrast to volume contraction, which leads to worse generalization performance.

## M Additional experiments

### M.1 Experiments on text classification

Table 13: Text classification for THUCNews .

| Method | FastText | FastText(dropout) | TextRNN | TextRNN(dropout) | TextCNN | TextCNN(dropout) |
|---|---|---|---|---|---|---|
| Sampling | 0.04±0.02 | **0.08±0.02** | 0.03±0.02 | **0.06±0.02** | 0.05±0.02 | **0.09±0.02** |
| Laplace | 0.0366 | **0.0831** | 0.0126 | **0.0458** | 0.0453 | **0.0850** |
| Train Loss | 0.01 | 0.08 | 0.04 | 0.07 | 0.02 | 0.04 |
| Test Loss | 0.66 | 0.25 | 0.46 | 0.29 | 0.61 | 0.42 |
| Loss Gap | 0.65 | **0.17** | 0.42 | **0.22** | 0.59 | **0.38** |
| Train Acc | 0.9958 | 0.9844 | 0.9841 | 0.9751 | 0.9944 | 0.9899 |
| Test Acc | 0.9011 | 0.9217 | 0.9037 | 0.9102 | 0.9022 | 0.9129 |
| Acc Gap | 0.0947 | **0.0627** | 0.0804 | **0.0649** | 0.0922 | **0.0770** |

To make our empirical results more robust, we also implement FastText (Bojanowski et al., 2017; Joulin et al., 2016; 2017), TextRNN (Liu et al., 2016) and TextCNN (Kim, 2014) to classify THUCNews data set (Sun et al., 2016) (180000 training data, 10000 validation data, 10000 test data) with and without dropout. The results of these text classification tasks in Table 13 also indicate that dropout noise can narrow generalization gap and expand the weight volume.

### M.2 Complexity measures with average sign-error

Table 14: Average sign-error of complexity measures on CIFAR-10.

| Complexity Measure | VGG GG | AlexNet GG |
|---|---|---|
| Frob Distance | 0.5374 | 0.5560 |
| Spectral Norm | 0.5045 | 0.6783 |
| Parameter Norm | 0.4817 | 0.6656 |
| Path Norm | 0.4852 | 0.4982 |
| Sharpness $\alpha'$ | 0.5068 | 0.6168 |
| PAC-Sharpness | 0.5079 | 0.4434 |
| **PAC-S(Laplace)** | **0.3763** | **0.3247** |
| **PAC-S(Sampling)** | 0.3781 | 0.3469 |

Figure 8: Some weight distributions of VGG 16 on CIFAR-10, red line is the true Gaussian, blue bar is the distribution of weights.

We also use average sign-error ($\in [0, 1]$) from Dziugaite et al. (2020b) to measure the effectiveness of complexity measures across all models. The smaller average sign-error means the better performance of complexity measures. Table 14 shows that our new complexity measures can also get better performances under the measure of average sign-error.

## M.3    Distribution of weights

Gaussian distributed weights are commonly assumed in the analysis of DNN models under the PAC-Bayesian framework. To verify its rationality, we conducted a new experiment in Figure 8. We collect weight samples with sampling method and the empirical results show that these random weights (blue bar) approximately obey a Gaussian (red line).

