# OpenReview forum: "Weight Expansion: A New Perspective on Dropout and Generalization"
_TMLR — Accepted by TMLR_

### Review · Reviewer_6GSD · 2022-06-01

**Summary Of Contributions:**

This paper presents a new explanation for how the dropout training algorithm improves generalization of neural network models. A quantity called "weight volume" --- defined as the normalized determininant of the weight covariance matrix --- is connected to generalization using the PAC-Bayes framework, and to dropout training by analyzing the correllation between gradients during dropout training. Experiments show correllations between dropout, weight volume, and generalization performance, performed with convolutional neural networks on CIFAR-10 and CIFAR-100. This new perspective suggests other forms of regularization that could be similar or better to dropout.

**Broader Impact Concerns:**

None.

**Requested Changes:**

- Perform a grammar check. There are some typos in the related works section.

**Strengths And Weaknesses:**

Strengths:
- The paper is well-written and organized. Clear hypotheses are presented, a theoretical justification is provided, and experiments are used to test the hypotheses.
- Experiments are systematic and well organized. Experiments are not on state-of-the-art models, but reasonable for this type of experiment.
- The idea of having a measureable quantity that would help predict the generalization ability of a model is quite appealing from a practical perpsective.

Weaknesses:
- The theoretical arguments rely on a long list of approximations and assumptions. The authors do a nice job of referring to other papers in the literature where these assumptions are used, but don't discuss the limits of these assumptions and where they break down.
- One simple example where it seems the theory might break down is in the analysis of dropout's connection to weight volume. When the dropout probabability q goes to 1, the gradient correlation goes to 0, but the weights themselves become highly correlated within a layer; the columns of W_l will converge to the same value, which will depend on the weights in the other layers. This scenario is not even unrealistic, as practicioners sometimes use high dropout rates in practice (e.g. 0.9), while the experiments in the paper only explore dropout rates up to 0.45.
- Marginal experimental improvement of proposed algorithm over standard dropout.

---

> ### Author Response · Authors · 2022-07-06
> **Response to Reviewer 6GSD**
>
> Dear Reviewer 6GSD,
>
> We appreciate your positive comments and are pleased that you recognize the novelty and originality of our contribution. In the following, we will address your questions in order.
>
> **For related work**
>
> According to your suggestion, we have re-organized the section of related work as 4 parts: 6.1 Related work about dropout; 6.2 Related work about other generalization factors; 6.3 Related work about PAC-Bayes; 6.4 Weight expansion and flatness. Please find Section 6 in the updated manuscript (re-written parts are marked in blue).
>
> **For the limits and the example that dropout probability $q$ goes to 1**
>
> Thank you for raising this interesting question.
> First, we would like to clarify the meaning of dropout rate $q$.
> For a theoretical dropout model, we just consider the case that the model is only updated by normal gradient during training.
> In this case, when $q$ goes to 1, it means there is no update during training. Thus the posterior Q is the same as prior P, so that the correlation is zero because of the i.i.d. assumption of P.
> In this case, the weight volume will be the largest (because it is the initial uncorrelated random weights) and the generalization gap is close to 0.
> If we understand incorrectly, please correct us.
>
>
> Nevertheless, we admit that there is a problem when dropout rate is large ($i.e.,$ close to 1), as it will seriously reduce the training accuracy ($e.g.,$ on VGG16). Although in this case the generalization gap will be small and weight volume will be large, the performance of the model is so bad that the dropout does not actually improve the training.  Thus, we only consider suitable dropout rate ($[0, 0.45]$) in our experiments. We have added this limitation to the final paragraph of Section 5.1.
>
> We hope our responses answer your questions and our revised manuscript is more clear. If you have further questions, please feel free to let us know.
>
> Best regards,
>
> Authors

---

### Review · Reviewer_yQtZ · 2022-06-09

**Summary Of Contributions:**

The paper studies dropout's effects on generalization. It introduces a notion of weight volume, and shows that under a particular perturbation of the weights, weight volume under dropout is preserved due to correlations among the weights being reduced over a single gradient step.

**Broader Impact Concerns:**

No concerns.

**Requested Changes:**

Critical changes based on the comments above:

 - clarifying the connections to flatness; add related work;
 - list all evidence toward supporting the two hypothesis presented in the introduction
- list assumptions for theoretical results clearly; clarify wording around one-step vs multi-step analysis
- add Section 5.2 extended evaluation based on two main competing methods
- re-write conclusion section;
 - respond to questions (1) and (2) above to prove correctness of the empirical evaluation of weight volume as a generalization measure.

Changes that would strengthen the paper:
- evaluate hypothesis one by perturbing weight volume in different ways. E.g., what if after optimization I optimized for the maximum weight  volume further, would that increase generalization?

**Strengths And Weaknesses:**

Overall, I enjoyed reading the paper and found it fairly interesting. One of my main concerns is that under local perturbations of the weights (such as Gaussian), weight volume is basically equivalent to flatness. To my understanding, weight volume is a particular way to measure flatness. Since the authors do not clearly state these connections, the novelty of the idea seems to be a bit overrepresented in the paper. I think there should be a separate paragraph explaining how weight volume and flatness connect, and the relevant citations need to be discussed (of which there should be quite a significant number).

Presentation: parts of the paper are fairly clearly written, parts are hard to follow (or make little sense, see below).

All theorem/lemma statements seem somewhat informal. The assumptions should be clearly listed in lemma statements, or listed prior to the statement and numbered, so that the theorem statement could refer to the assumptions. For example, it was too obscured in my opinion that P and Q share the same diagonal for obtaining the result in Lemma 3.1. I only realized after looking at the proof in the appendix. In addition, Lemma’s 3.1 and 3.2 present one-step analysis of correlations of the weights. The authors should make sure that the claims made in various discussions are accurate and point out that the analysis is one-step, or present multi-step analysis.

Section 7 (Conclusion and future work) is very poorly written. The section is full of vacuous statements, and reads more like a space filler. It is hard to follow the logical structure (if there is structure). What’s the connection to adversarial generalization? What exactly is adversarial generalization? Why would the optimal solution not be practical? It was just said it is intractable, so it seems one cannot even go searching for this optimal solution, practical or not.

Claims: i personally would prefer to see all the claims made in the paper and the respective evidence listed somewhere (with a precise description of the evidence, e.g., distinguishing between one-step and multi-step analysis, the extent of the experimental procedure, etc.)

Experiments: the authors present two hypotheses that relate generalization, dropout, and weight expansion. They then present some empirical evidence that the quantities of interest are correlated. However, to truly test the hypothesis, some interventions changing the quantities of interest are needed. Let’s, for example, consider Hypothesis 1: “weight expansion reduces the generalization error”. In my opinion, to test such a hypothesis, one would need to think of experiments that could alter the weight expansion, and not through dropout only. I know the authors consider disentanglement noise, but that seems to be directly related to training neural networks essentially for perturbation robustness.

In Section 5.2 the authors claim to follow Jiang et al and Dziugaite et al for measuring correlations between generalization measures and generalization. Note, however, that the two mentioned papers propose different ways to evaluate the relationship between generalization and a generalization measure: one considers an average loss, the other considers a distributionally-robust loss. The authors should more carefully check both pieces of work, and have an appropriate discussion why they choose one over the other (or measure both).

Regarding the evaluation of weight volume as a generalization measure:
1) how is sigma for the prior chosen?  This detail may be mentioned in the paper but was not easy to find, and is important for the correctness of the evaluation of weight volume as a generalization measure (at least if the measure is justified through a PAC-Bayes bound).
2) Is generalization measured of a perturbed or unperturbed classifier?

Missing citations:  Huang et al ‘20 also used block-diagonal structure of the covariance matrix; McAllister ‘13 “A PAC-Bayesian tutorial with a dropout bound” – different from the submission, as the noise considered is diagonal Gaussian, and thus the weight volume part does not appear.

---

> ### Author Response · Authors · 2022-07-06
> **Response to Reviewer yQtZ (Part 2/2)**
>
> **7. Evaluate hypothesis one by perturbing weight volume in different ways. E.g., what if after optimization I optimized for the maximum weight volume further, would that increase generalization?**
>
> Thank you for the insightful suggestion.
> Actually we have tried to verify hypothesis (1) in two other ways as follows.
>
> 1. We tried to optimize weight volume directly, but can not find an effective way to achieve it. In our experiments, weight volume is only an estimate, which makes it difficult to be operated directly. Thus, we chose to use disentanglement noise to optimize it in an indirect way.
>
> 2. We tried to further optimize weight volume after model converges, but it turned out that such optimization would damage training accuracy. Although further optimization could narrow generalization gap, it also damages test accuracy. For this reason, we did not adopt this method.
>
> **8. Missing citations**
>
> We added the discussion of (McAllister ‘13) in Section 6.3 and the discussion of (Huang et al ‘20) in Section 6.2.
>
> We hope our responses answer your questions and our revised manuscript is clearer. If you have further questions, please feel free to let us know.
>
> Best Regards,
>
> Authors

---

> ### Author Response · Authors · 2022-07-06
> **Response to Reviewer yQtZ (Part 1/2)**
>
> Dear Reviewer yQtZ,
>
> We appreciate your positive and detailed comments and are pleased that you enjoyed reading our manuscript. In the following, we will address the questions in order.
>
> **1. Clarifying the connections to flatness; add related work.**
>
> According to your suggestion, we have re-organized the section of related work as 4 parts: 6.1 Related work about dropout; 6.2 Related work about other generalization factors; 6.3 Related work about PAC-Bayes; 6.4 Weight expansion and flatness. In particular, we clarified the connection between weight volume and flatness in Section 6.4. Please find Section 6 in the updated manuscript (re-written parts are marked in blue).
>
> **2. List all evidence toward supporting the two hypotheses presented in the introduction**
>
> Following your suggestion, we have re-written some paragraphs in Introduction to present the supporting evidence of two hypotheses. Please refer to the last 4 paragraphs in Section 1 for details.
>
> **3. List assumptions for theoretical results clearly; clarify wording around one-step vs multi-step analysis**
>
> According to your suggestion, we added the assumptions into the lemmas to make them more explicit, as shown in the main text. In addition, with respect to one-step and multi-step analysis,
> we re-wrote Section 3 with the following points emphasized.
>
> 1. Lemma 3.1 describes the KL divergence between weight prior (before training) and posterior (after training) under the PAC-Bayes framework, in which we regard the whole training as a single multi-step procedure.
>
> 2. When it comes to the correlation, it is too complicated for a tractable analysis during the whole training procedure, which is a highly dynamic process. For this reason, Lemmas 3.2 and 3.3 adopt one-step theoretical arguments (one mini-batch update) to indicate how dropout reduces correlation in each step. With the same pre-condition, we prove that dropout noise can reduce the correlation among weights.
>
> 3. Whilst it is challenging to obtain the multi-step analysis during the whole training, we demonstrate in Figure 2 empirically that dropout reduces correlation in the whole (multi-step) training procedure.
>
> **4. Add Section 5.2 extended evaluation based on two main competing methods.**
>
> We agree that the measurement in (Dziugaite et al 2020) is different from (Jiang et al 2019), and rectified the corresponding statements in the revised manuscript. Following your suggestion, we extended evaluation based on those two methods. We chose the measurement from (Jiang et al 2019) mainly because it can be quickly deployed by us based on the TensorFlow implementation in (Jiang et al 2019; Jiang et al NeurIPS 2020 competition). In addition, we have added the experiments with the average sign-error measurement of (Dziugaite et al 2020) in Appendix M.2, and it turns out that the empirical results can also support our hypotheses.
>
> **5. Re-write conclusion section.**
>
> Thank you for raising these issues. In the revised version, we re-organized the conclusion in a logical way and made the statements more clearly.
>
> **6. Respond to questions (1) and (2) above to prove correctness of the empirical evaluation of weight volume as a generalization measure.**
>
> **(1) how is sigma for the prior chosen?**
>
> Following previous works, we use the sharpness PAC-Bayes (Jiang et al 2019) (also known as flatness PAC-Bayes in Dziugaite et al 2020) in this paper. The sigma is estimated by perturbed Gaussian noise (see details in Appendix K.2 Eq. (29)). To make it clearer, we have emphasized it in Section 5.2.
>
> **(2) Is generalization measured of a perturbed or unperturbed classifier?**
>
> The perturbed models are used to estimate weight volume (by using sampling method) and sharpness PAC-Bayes sigma. Then, the complexity measures with these estimates are further used to predict the generalization of unperturbed models. This point is now highlighted in Section 5.2.

---

> > ### Comment · Reviewer_yQtZ · 2022-07-24
> > **Other citation issues**
> >
> > Thank you for addressing most of my comments.
> >
> > Reading through your changes, I noticed that Section 6.4 on flatness and generalization is still missing a citation to Neyshabur et al and Dziugaite and Roy, where the authors connect flatness to generalization via pac-bayes (i.e., not just empirically show that flatness and generalization correlate, like in Keskar et al). I hope the authors will make appropriate edits.
> >
> > That said, I also expected to see a more direct comparison between weight volume and flatness. E.g., assuming a quadratic approximation of the loss holds, how does weight volume relate to the curvature via hessian eigenvalues?

---

> > > ### Author Response · Authors · 2022-07-25
> > > **Response to 'Other citation issues'**
> > >
> > > Thank you for your comment, it really helps us make clear the relationship between weight volume and flatness. We have updated Section 6.4 in the revised manuscript, for your convenience, the added paragraph is as below:
> > >
> > > Of most relevance is the work by Dziugaite & Roy (2017); Neyshabur et al. (2017; 2018a), which relates a new concept of expected flatness over random weights to the PAC-Bayesian framework. By relaxing their i.i.d. assumption of random weights, our proposed concept of weight volume generalizes the expected flatness (that considers solely the variance of random weights) with weight correlation taken into account (that considers the weight covariance). Specifically, weight volume measures the expected flatness using the determinant of normalized covariance matrix of random weights. That is, a larger weight volume means random weights lie in a flatter local minimum (obeying a more spherical distribution), which can lead to better generalization. In particular, assuming a quadratic approximation of the loss holds, say Laplace approximation, weight volume approximately represents the inverse of the product of  all eigenvalues of the Hessian (without normalization). That is, $\mathrm{vol(\mathbf{w})}\approx\prod_i \frac{1}{\lambda_i}$, where $\lambda_i$ is the $i$-th eigenvalue of the Hessian matrix $H$. Not only does this confirm the connection between weight volume and sharp/flat minima (i.e., large/small eigenvalues) of the Hessian, but also provides another, perhaps more comprehensive, view of flatness with respect to the whole spectra of the Hessian.
> > >
> > > If you have further comments, please feel free to let us know. We are happy to hear from you.
> > >
> > > Best Regards,
> > >
> > > Authors

---

### Review · Reviewer_YqAv · 2022-06-25

**Summary Of Contributions:**

This paper studies the connection between dropout and generalization with a new concept: weight expansion. In particular, the authors show that the PAC-Bayesian generalization error is related to the weight volume, and dropout can lead to a large weight volume. Based on this finding, this paper developed two approaches to approximately calculate the weight volume. Finally, the experimental results confirm that weight volume is a good indicator of generalization performance.

**Requested Changes:**

1. In this paper, the weight is assumed to follow a Gaussian distribution, and then the generalization bound is obtained. Is this true for real-world applications? It would be good if the authors could conduct experiments to show the distribution of the weight of some DNNs.
2. The proposed two approaches for computing the weight volume look slow. It would be good if the authors could show the consumed time.

**Strengths And Weaknesses:**


Pros:
1. The idea is novel. It builds the connection between the generalization performance and the weight volume.
2. The methods for calculating the weight volume are effective.
3. The experimental results confirm the performance of the proposed methods.
4. The writing is easy to follow.

Cons:
1. In this paper, the weight is assumed to follow a Gaussian distribution, and then the generalization bound is obtained. Is this true for real-world applications? It would be good if the authors could conduct experiments to show the distribution of the weight of some DNNs.
2. The proposed two approaches for computing the weight volume look slow. It would be good if the authors could show the consumed time.

---

> ### Author Response · Authors · 2022-07-06
> **Response to Reviewer YqAv**
>
> Dear Reviewer YqAv,
>
> We appreciate your positive comments and are pleased that you recognize the novelty and originality of our contribution. In the following, we will address your questions in order.
>
> **1. For the first suggestion, the true distribution of DNNs.**
>
> Gaussian distributed weights are commonly assumed in the analysis of DNN models under the PAC-Bayesian framework, as in [1-4].
> To verify its rationality, we conducted a new experiment, and the results are presented in Appendix M.3.
> In the experiment, we collected samples with sampling methods in Figure 8.
> The empirical results show that these random weights (blue bar) approximately obey a Gaussian (red line).
>
> **2. For the second suggestion, show the consumed time of weight volume estimation.**
>
> The consumed time of estimating weight volume is shown in Section 5 now.
> For each VGG model or AlexNet-like models, it takes about 1,500s and 300s to (collect samples and) estimate weight volume for the sampling method and the Laplace approximation, respectively .
> Compared with the consumed time of computing other complexity measures and training, the time consumption of estimating weight volume may be negligible.
>
> We hope our responses answer your questions and our revised manuscript is more clear. If you have further questions, please feel free to let us know.
>
> [1] Gintare Karolina Dziugaite and Daniel M Roy. Computing nonvacuous generalization bounds for deep (stochas-tic) neural networks with many more parameters than training data. UAI, 2017.
>
> [2] Behnam Neyshabur, Srinadh Bhojanapalli, David Mcallester, and Nati Srebro. Exploring generalization in deep learning. Advances in Neural Information Processing Systems, 30:5947–5956, 2017.
>
> [3] Yiding Jiang, Behnam Neyshabur, Hossein Mobahi, Dilip Krishnan, and Samy Bengio. Fantastic generalization measures and where to find them. International Conference on Learning Representations. 2019.
>
> [4] Gintare Karolina Dziugaite, Alexandre Drouin, Brady Neal, Nitarshan Rajkumar, Ethan Caballero, Linbo Wang, Ioannis Mitliagkas, and Daniel M Roy. In search of robust measures of generalization. Advances in Neural Information Processing Systems, 33, 2020b.
>
> Best regards,
>
> Authors

---

### Author Response · Authors · 2022-07-06
**Revised manuscript**

Dear Action Editor and Reviewers,

Thank you for your time and constructive feedback! We have uploaded a revision of the manuscript according to your comments, which contains the following modifications (re-written parts are marked in blue):

1. Section 1 Introduction: we have re-written some paragraphs in Introduction to present the supporting evidence of two hypotheses.

2. Section 3 Dropout and weight expansion: we added the assumptions into the lemmas to make them more explicit, in addition, we re-wrote some paragraphs with one-step and multi-step analysis.

3. Section 5 Experiments: (1) we show the consumed time of estimating weight volume; (2) we rectified the statement of measurement for complexity measures; (3) we show how to estimate sharpness PAC-Baye sigma; (4) we add the limitations of our methods.

4. Section 6 Discussion: we have re-organized this section as 4 parts: 6.1 Related work about dropout; 6.2 Related work about other generalization factors; 6.3 Related work about PAC-Bayes; 6.4 Weight expansion and flatness. In particular, we clarified the connection between weight volume and flatness in Section 6.4.

5. Section 7 Conclusion and future work: we re-organized the conclusion in a logical way and made the statements more clearly.

6. Appendix M.2: we show the complexity measure experiments with the average sign-error measurement from Dziugaite et al. (2020b).

7. Appendix M.3: we show the true distribution of some weights with sampling method.

Best Regards,

Authors

---

### Decision · Action_Editors · 2022-08-14

**Recommendation:** Accept with minor revision

**Comment:**

This work explored a connection between weight expansion (measured by the determinant of normalized weight covariance) and dropout in neural network training. In particular, the authors demonstrated that weight expansion is an effective means of increasing generalization in a PAC-Bayesian setting, and argued that dropout, among other methods, indeed led to weight expansion. The authors performed some initial theoretical analysis and conducted extensive experiments to validate their claims.

All three reviewers found this work interesting in the sense of providing a new perspective on the widely adopted dropout. Some questions on writing, related work, and assumptions were posed, which the authors satisfactorily addressed in the response. In the end, all reviewers recommended acceptance, which the AE concurs.

Some minor suggestions for the final version:
- Please consider abandoning the abbreviation "do" for dropout, since "do" already has an agreed-on meaning in causality. Using the tilde notation (e.g. $\tilde\eta$) probably suffices.
- In the newly added Lemma 3.2 (as well as the text above it), ${\Delta^{do}}'$ was never defined. The assumption on the same distribution of ${\Delta^{do}}'$ seems a bit strong and falls into the single-step analysis pointed out by the reviewers.
- Representing dropout as $(1+\eta)\odot a$ seems a bit awkward. Simply let $\tilde X = \frac{\delta}{1-q} X$ (with $X$ representing any quantity of interest), where $\delta \sim \mathrm{Bernoulli}(1-q)$, i.e., $\delta = 1$ with prob. $1-q$, would be more direct. For instance, the proof of Lemma 3.2 can be largely trivialized: $E(\tilde X \tilde Z) = E(\frac{\delta_X}{1-q} X \frac{\delta_Z}{1-q}  Z) = E(XZ)$, by conditioning on $X$ and $Z$ and taking expectation on the independent Bernoulli $\delta_X$ and $\delta_Z$. Similarly for $\mathrm{Var}(\tilde X)$ as well as Lemma 3.3.

---

> ### Author Response · Authors · 2022-09-01
> **Camera-Ready Version**
>
> Dear Action Editor,
>
> We appreciate your helpful suggestions and have updated the manuscript according to your suggestions 1 and 3.
>
> Follow your suggestion 2, we have emphasized the assumption that $\Delta$ and $\tilde{\Delta}'$ are i.i.d. above Lem. 3.2 to make the definition of $\tilde{\Delta}'$ clear.
> However, for the current work, we have to use this assumption to make the theoretical analyses in Lems. 3.2 and 3.3 hold.
> Nevertheless, we may relax this assumption to study dropout in our future work.
>
> Please check our camera-ready version and feel free to let us know if there is any question.
>
> Best Regards,
> Authors